# Influence of the vessel wall geometry on the wall-induced migration of red blood cells

**Ying Zhang**  *, **Thomas G. Fai**

Department of Mathematics, Brandeis University, Waltham, Massachusetts, United States of America

* yingzhang@brandeis.edu

## Abstract

The geometry of the blood vessel wall plays a regulatory role on the motion of red blood cells (RBCs). The overall topography of the vessel wall depends on many features, among which the endothelial lining of the endothelial surface layer (ESL) is an important one. The endothelial lining of vessel walls presents a large surface area for exchanging materials between blood and tissues. The ESL plays a critical role in regulating vascular permeability, hindering leukocyte adhesion as well as inhibiting coagulation during inflammation. Changes in the ESL structure are believed to cause vascular hyperpermeability and entrap immune cells during sepsis, which could significantly alter the vessel wall geometry and disturb interactions between RBCs and the vessel wall, including the wall-induced migration of RBCs and the thickening of a cell-free layer. To investigate the influence of the vessel wall geometry particularly changed by the ESL under various pathological conditions, such as sepsis, on the motion of RBCs, we developed two models to represent the ESL using the immersed boundary method in two dimensions. In particular, we used simulations to study how the lift force and drag force on a RBC near the vessel wall vary with different wall thickness, spatial variation, and permeability associated with changes in the vessel wall geometry. We find that the spatial variation of the wall has a significant effect on the wall-induced migration of the RBC for a high permeability, and that the wall-induced migration is significantly inhibited as the vessel diameter is increased.

## Author summary

The interaction between the blood vessel walls and flowing blood cells plays a pivotal role in maintaining the spatially non-uniform distribution of blood cells in the vessel lumen. However, it is not yet well-understood how changes in the vessel wall geometry caused by the disturbances in the endothelial surface layer (ESL) affect the microcirculation. In this work, we show that the wall-induced migration of red blood cells (RBCs) may be significantly affected by changes in the wall thickness and spatial variation caused by changes in the ESL. By combining our model with medical images showing changes that occur during sepsis, we demonstrate how the vessel wall geometry affects the development of the cell-free layer (CFL) in disease states. Our work highlights the influence of the vessel wall

**Data Availability Statement:** All relevant data are within the paper and its Supporting information files. An open source code repository containing MATLAB code used for producing the data can be

**Funding:** This work was supported by National Science Foundation grant DMS-1913093 to TGF. The funders had no role in study design, data collection and analysis, decision to publish, or preparation of the manuscript.

geometry induced by changes in the ESL on the wall-induced migration of RBCs and the formation of CFL.

This is a *PLOS Computational Biology* Methods paper.

## Introduction

Blood flow is a key determinant in numerous pathologies such as tumor vascularization [1], sickle cell anemia [2], and atherosclerosis [3]. Along with plasma, red blood cells (RBCs) are major components of blood, occupying 40% − 45% of its volume [4]. At rest, a healthy RBC has a biconcave shape with a diameter of 8 $\mu$m and a width of 2 $\mu$m and is highly deformable, which allows them to pass through capillary vessels with a diameter as small as 2.7 $\mu$m [5]. Due to the high occupancy of RBCs, the blood flow is, therefore, greatly influenced by the dynamics of RBCs.

Fluid-mediated interactions between circulating RBCs and vessel walls lead to a non-uniform distribution of cells and spatially dependent resistance to flow. Physical features of the vessel wall, such as the wall thickness and permeability, are found to be critical factors that contribute to different spatial profiles of RBCs. *In vitro* experiments of blood flow in narrow glass tubes reveal the cross-stream migration of RBCs, a phenomenon that is often referred to as wall-induced migration. This migration is due to the wall-induced lift force generated on deformable cells by the non-uniform shear near vessel wall [6]. For deformable objects such as RBCs the presence of a wall near the cells generates asymmetrical forces on the cells, leading to their lateral migration [7–9]. Further analyses of deformable vesicles have revealed that the generation of lift forces is caused by asymmetry of vesicle shape or orientation [8, 9]. One important physical phenomena resulting from the wall-induced migration of RBCs is the formation of a cell-depleted layer or cell-free layer (CFL) near the wall [10–12]. Understanding CFL formation in the microcirculation is of significant interest for the following three reasons: (1) the CFL contributes to the rheological properties of blood in arterioles and venules [13], (2) the CFL is shown to modulate the nitric oxide scavenging effects by RBCs in arterioles [14], and (3) it enhances plasma skimming in terminal arterioles, which can increase the heterogeneity of the RBC distribution in capillaries [15–17]. Previous studies using a tube to represent the blood vessel suggest that the formation of CFL depends on many factors such as vessel diameter and hematocrit [18–20]. In particular, when the hematocrit is held constant, the CFL thickness increases as the vessel diameter is increased and the mean blood velocity is faster. On the other hand, when the vessel diameter is fixed, the CFL thickness increases as the hematocrit is decreased and the mean blood velocity is faster. As the motion of RBCs and the formation of CFL are largely controlled by the overall topography of the vessel wall, understanding changes in the wall geometry is important. One special case where the wall geometry can be changed is through changes occur in the endothelial surface layer (ESL). The presence of the ESL could significantly alter the vessel wall geometry under different pathological conditions and it is believed to be a contributing factor to the substantial increase in flow resistance [21] and potentially affect the formation of CFL. The ESL is made of two semi-distinct layers of membrane-bound macromolecules. The inner part comprises a thin layer (0.05 − 0.4 $\mu$m) of molecules dominated by glycoproteins and

proteoglycans that are anchored to the endothelial cells. This layer has been observed to have a quasi-periodic matrix structure [22] with brush-like configurations of fibers [23, 24]. The outer layer consists of a complex array of soluble plasma proteins and glycosaminoglycans that extends approximately 0.5 $\mu$m from the inner layer [25]. Upon interacting with RBCs, the whole layer tends to resist compression and flattening [21, 26].

Over the past decade, several theoretical models have been developed to further investigate the effects of the ESL on blood flow and the mechanics of RBC motion. Previous theoretical studies have considered the motion of rigid spherical particles (representing RBCs) through cylindrical tubes lined with a porous wall layer (representing the ESL) [27, 28] using lubrication theory and revealed that the shear stress on the solid sphere is greatly reduced with the presence of the porous layer. To take into consideration the effect of RBC shape and deformability, more detailed models have been developed. In [29] the authors analyze how ESL properties influence the partitioning of RBCs at vessel bifurcations. In [21, 30], a theoretical model is built based on [27] to analyze the motion and deformation of RBCs in a capillary lined with ESL and to predict the effect of ESL on flow resistance and hematocrit. The RBC is modeled as an axisymmetric viscoelastic membrane containing an incompressible viscous fluid. Through analyzing the Fahraeus effect, which states that the hematocrit decreases with decreasing capillary diameter, and flow resistance, it has been shown that the presence of the ESL accentuates the Fahraeus effect and can substantially increase flow resistance in capillaries. Modeling approaches other than using the lubrication theory have also been used. In [31], a simplified two-dimensional model is used to examine the effect of the ESL, considered as a porous medium, on lateral migration of RBCs when the cell is initialized close to the layer in a parallel-sided channel. The RBC is represented as a set of viscoelastic elements on the perimeter of cell with a set of viscous elements in the interior. The flow of plasma is modeled using the Brinkman approximation. By analyzing the deformation, motion of the RBC and the fluid stresses acting on the RBC, the model predicts that the tendency of the RBC to migrate away from the ESL decreases as the layer becomes more permeable and would lead to a decreased width of CFL. In a more recent study, the interaction between the ESL and the RBC is studied using dissipative particle dynamics [32]. A microscopic description of the ESL is considered. The ESL is described as chains by a bead-string model. Each chain consists of beads connected by elastic springs with bending rigidity. At equilibrium springs are assumed to have a specified resting length and the angle among three neighboring beads is assumed to be $\pi$. Through examining various chain layout, chain length, RBC velocities, the model suggests that RBCs drive the ESL deformation via the near-field flow, whereas marginal propulsion of RBCs by ESL is observed.

While a number of models and numerical methods have been used to study the effect of ESL on blood flow and motion of RBCs, how it affects the wall-induced migration of RBCs and the formation of CFL via changing the vessel wall geometry remains unclear. In this work, we followed the approach used in previous studies [33, 34] to develop a two-dimensional computational model to study the effect of the vessel wall geometry associated with changes in the ESL on the wall-induced migration of RBCs and in particular the formation of CFL during sepsis. Our computational model differs other existing models in two ways: (1) interaction among blood flow, RBCs, and vessel wall is coupled using an immersed boundary method [35] and (2) permeability of the vessel wall associated with the ESL is imposed through Darcy's law [36]. The computational model used here offers the flexibility of handling complex vessel wall geometries, which is beneficial for incorporating experimental data. To understand the influence of the vessel wall geometry on the motion of the RBC near the wall under different ESL conditions, we developed two models that capture the geometry of the wall at different levels of resolution. In the macroscopic ESL model, the topography of the wall is coarse-grained into

a smooth wave (Fig 1A), whereas in the microscopic ESL model, the brush-like structure of the ESL is applied to represent the wall geometry explicitly (Fig 1B). An open source code repository containing MATLAB code used for the simulations may be found at https://github.com/phzhang0616/RBC-ESL_IB_Simulation.git. In Appendix C in S1 Text we demonstrate the validity of our model by examining two scenarios corresponding to impermeable ESL layers and permeable layers in [31] and compare our results to those reported in [31].

We first used our models to study computationally how the motion of a *single* RBC varies as the spatial variation, thickness, and permeability of the vessel wall is changed in a capillary, considered as a rectangle. The motion of the cell is determined by the force and the mobility. The coefficients of the mobility tensor depend on the distance to the vessel wall (see Appendix F in S1 Text) [37–41]. The force consists of components parallel and perpendicular to the vessel wall, and we quantify the migration motion of the RBC via analyzing the lift force (i.e. force perpendicular to incoming flow) and the drag (i.e. force induced by the wall that is parallel to the opposite direction of incoming flow) experienced by the cell. In particular, the lift is used to assess the ability of the RBC to escape from the vessel wall. Lateral migration is an emergent behavior that involves the interactions between cell deformations and the surrounding flow. The shape asymmetry induced by local flow gradients is a direct consequence of the increasing resistance toward the no-slip wall, and the lift on the cell depends on these changes in shape [42]. Using the microscopic ESL model, we find that the wall-induced migration of the RBC is hindered as the density of bundles increases for a high permeability. In contrast, for a small permeability we find that the wall-induced migration of the RBC remains largely unchanged. We further show that the RBC is less likely to migrate away from the wall as the thickness is increased, and the permeability does not appear to have a strong influence on the wall-induced migration as the thickness varies. We reproduced these results using a more computationally efficient macroscopic ESL model, in which the wall is described as a sinusoidal wave.

To understand the changes in microcirculation with vessel wall geometry associated with healthy and disrupted ESLs in a realistic setting, we extract the vessel wall geometry from medical images provided in [43], taken under a healthy and a septic scenario. Inflammation in general and sepsis in particular result in changes in the ESL mainly in its permeability and structural changes caused by shedding. Previous studies on mice reveal that mice lacking the TNF receptor 1 were resistant to heparanase and an enzyme that degrades heparan sulfate, which activates ESL degradation and thus decrease the permeability of the layer [44, 45]. Shedding of the ESL occurs in the presence of oxidants, hyperglycemia, cytokines, and bacterial

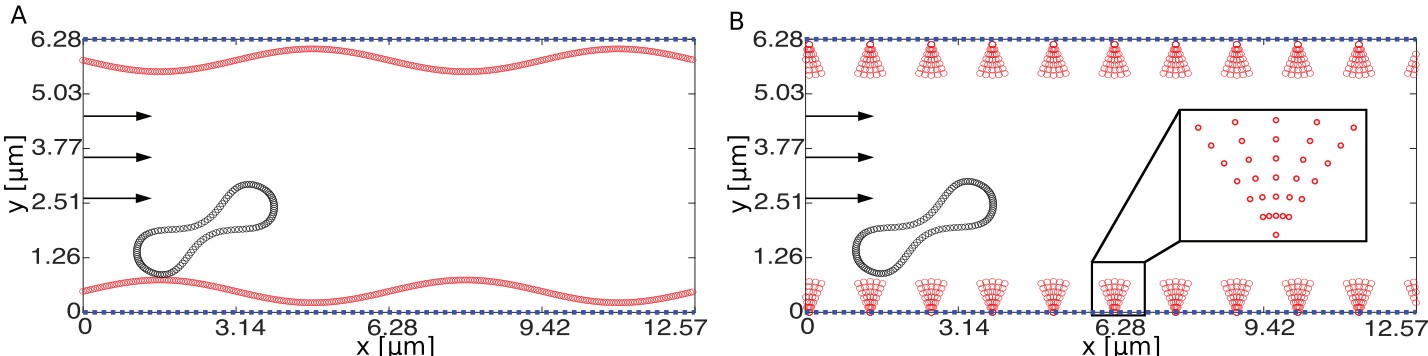

**Fig 1. An illustration of the vessel wall layout with a RBC at its initial position.** (A) the macroscopic ESL model and (B) the microscopic ESL model. In both panels the vessel wall layout is labeled in red and the RBC is labeled in black. The no-slip boundary on the top and the bottom of the channel, representing the vessel wall, is labeled using a blue dotted line. Arrows at the inlet indicate the direction of flow.

endotoxins, which is closely associated with sepsis. We note that while shedding of the ESL may lead to narrowed blood vessels via adhesion of blood constituents such as leukocytes, it is not actually a change in ESL thickness but rather the adhesion of leukocytes that causes the narrowing of the vessel. [46, 47]. To obtain insights about how changes in the vessel wall geometry induced by changes in the ESL in sepsis using data from experiments, we apply the macroscopic ESL model using the extracted vessel wall geometry and analyze the CFL thickness in simulations including only RBCs or both RBCs and leukocytes with different hematocrits. In all cases, we find that the fraction of the vessel diameter occupied by the CFL increases significantly in the septic scenario. The nonaxisymmetric nature of the CFL is preserved in the healthy and septic cases. Our work highlights the importance of the ESL-induced vessel wall geometry in studying microcirculation, and suggests a possible unexpected role for the ESL in controlling the formation of the CFL in different pathologies.

## Materials and methods

### Mathematical model of blood flow

We let $\Omega \subset \mathbb{R}^2$ be a bounded domain representing the blood vessel filled with plasma, which is incompressible, Newtonian, and contains immersed RBCs and vessel walls. In all the simulations, we assume $\Omega$ is a rectangle. We consider situations in capillaries where the Reynolds number is small allowing us to describe the dynamics of blood flow via the unsteady Stokes equation [48–50]

$$\mathrm{Re}\frac{\partial \boldsymbol{u}}{\partial t} + \nabla p = \Delta \boldsymbol{u} + \boldsymbol{f}$$
$$\nabla \cdot \boldsymbol{u} = 0,$$

$$(1)$$

equipped with boundary conditions as follows

$$\boldsymbol{u} = \boldsymbol{0} \quad \text{on the top and bottom of } \Omega,$$
$$\boldsymbol{u} \quad \text{periodic in the } x \text{ direction.}$$

$$(2)$$

In Eq 1 $\boldsymbol{u}(\boldsymbol{x}, t)$ denotes the fluid velocity, $p(\boldsymbol{x}, t)$ is the pressure, both defined in terms of the Eulerian coordinates, $\boldsymbol{x} = (x, y)$, and Re is the Reynolds number. The force, $\boldsymbol{f}(\boldsymbol{x}, t)$, applied to the blood is given by

$$\boldsymbol{f}(\boldsymbol{x}, t) = \boldsymbol{f}_{\mathrm{IB}}(\boldsymbol{x}, t) + \boldsymbol{f}_{\mathrm{Body}}(\boldsymbol{x}, t),$$

$$(3)$$

where $\boldsymbol{f}_{\mathrm{IB}}(\boldsymbol{x}, t)$ is the force induced by the presence of RBCs and ESLs (see Methods) and $\boldsymbol{f}_{\mathrm{Body}}(\boldsymbol{x}, t)$ is a body force given by

$$\boldsymbol{f}_{\mathrm{Body}}(\boldsymbol{x}, t) = (\sin(\pi y), 0)$$

$$(4)$$

to establish a sinusoidal flow profile as in [50]. In the absence of the immersed objects, this establishes a unidirectional flow in the domain.

### Elasticity model for the RBC membrane

We use a two-dimensional elastic spring model to describe the elasticity of the immersed RBCs, which is represented on a moving Lagrangian grid with coordinates $q \in [0, L_R]$ and the corresponding Eulerian coordinates at time $t$ are given by $\boldsymbol{X}(q, t)$. To describe the RBC membrane energy, we followed the approach in [34, 51–53]. We assume that the membrane Lagrangian points are connected by springs having total elastic energy for stretching/

compressing

$$E_{\text{spring}} = \frac{k_s}{2} \int_0^{L_R} \left( \left\| \frac{\partial \boldsymbol{X}(q,t)}{\partial q} \right\| - 1 \right)^2 dq, \tag{5}$$

and by torsional springs with total energy for bending

$$E_{\text{bend}} = \frac{k_b}{2} \int_0^{L_R} \left\| \frac{\partial^2 \boldsymbol{X}(q,t)}{\partial q^2} \right\|^2 dq, \tag{6}$$

The coefficients $k_s$ and $k_b$ are elastic stretching and bending constants respectively. When the RBC deforms, its surface area and volume remain relatively constant. To conserve the area of RBC in the model, we prescribe a penalty energy

$$E_{\text{area}} = \frac{k_a}{2} \left( A(t) - A_0 \right)^2, \tag{7}$$

where $k_a$ is an area-preserving constant, $A(t)$ is the area of the RBC at time $t$ and $A_0$ is the target area. The area-preserving constant $k_a$ is chosen so that in all simulations the maximum loss of area of the RBC is less than 2%. Note that as shown in [54] the standard immersed boundary method may suffer from poor area conservation in long-time simulations. It is, therefore, necessary to include this area conservation energy to preserve the total area and perimeter of all cells in our simulations. We further note that we do not impose a reference curvature as Eq 6 together with Eq 7 are suffice to obtain red-blood-cell-like shapes [51]. Additionally, although using Eq 6 does not yield the exact curvature, especially for large deformations, because of the compressibility of the membrane, the total arc length of the RBC in all the simulations differs from the reference arclength by less than 3%. The resulting Lagrangian force on the RBC membrane maybe be calculated via

$$\boldsymbol{F}_{\text{RBC}} = - \left[ \frac{\varrho E_{\text{spring}}}{\varrho \boldsymbol{X}} + \frac{\varrho E_{\text{bend}}}{\varrho \boldsymbol{X}} + \frac{\varrho E_{\text{area}}}{\varrho \boldsymbol{X}} \right], \tag{8}$$

where $\varrho / \varrho \boldsymbol{X}$ is the Fréchet derivative.

## Elasticity model for the vessel wall

A key consideration of the model is how the vessel wall geometry is influenced by changes in the ESL. Due to the lack of direct measurements of the mechanical properties of the ESL, we followed the information provided in previous studies [21, 55] to model the ESL as an elastic structure that is not advected with the blood flow. Observations of the restoration of ESL thickness within approximately one second following its compression by a passing white blood cell indicate that the layer is deformable with a finite resistance to compression [56]. Following this evidence, we modeled the wall as an elastic structure. To study the influence of the wall thickness and spatial variation on the near-wall motion of a RBC, we used curvilinear boundaries on a Lagrangian grid with coordinates $\hat{q} \in [0, N_E]$ to represent its structure. We assume that there are two walls on the top and bottom of the computational domain with the corresponding Eulerian coordinates given by $\hat{\boldsymbol{X}}_0(\hat{q}, t)$ and $\hat{\boldsymbol{X}}_L(\hat{q}, t)$ respectively. In the microscopic model, where the brush-like structure of the ESL is explicitly used to represent the vessel wall geometry, we described the wall as bundles of fibers (see Figs 1B and 2A) equipped an elastic energy that is defined identically as Eq 5. As the ESL is mostly immobile, the root of each fiber is tethered to tether points $\boldsymbol{Z}_0(t)$ and $\boldsymbol{Z}_L(t)$ at the domain boundary by stiff springs with spring

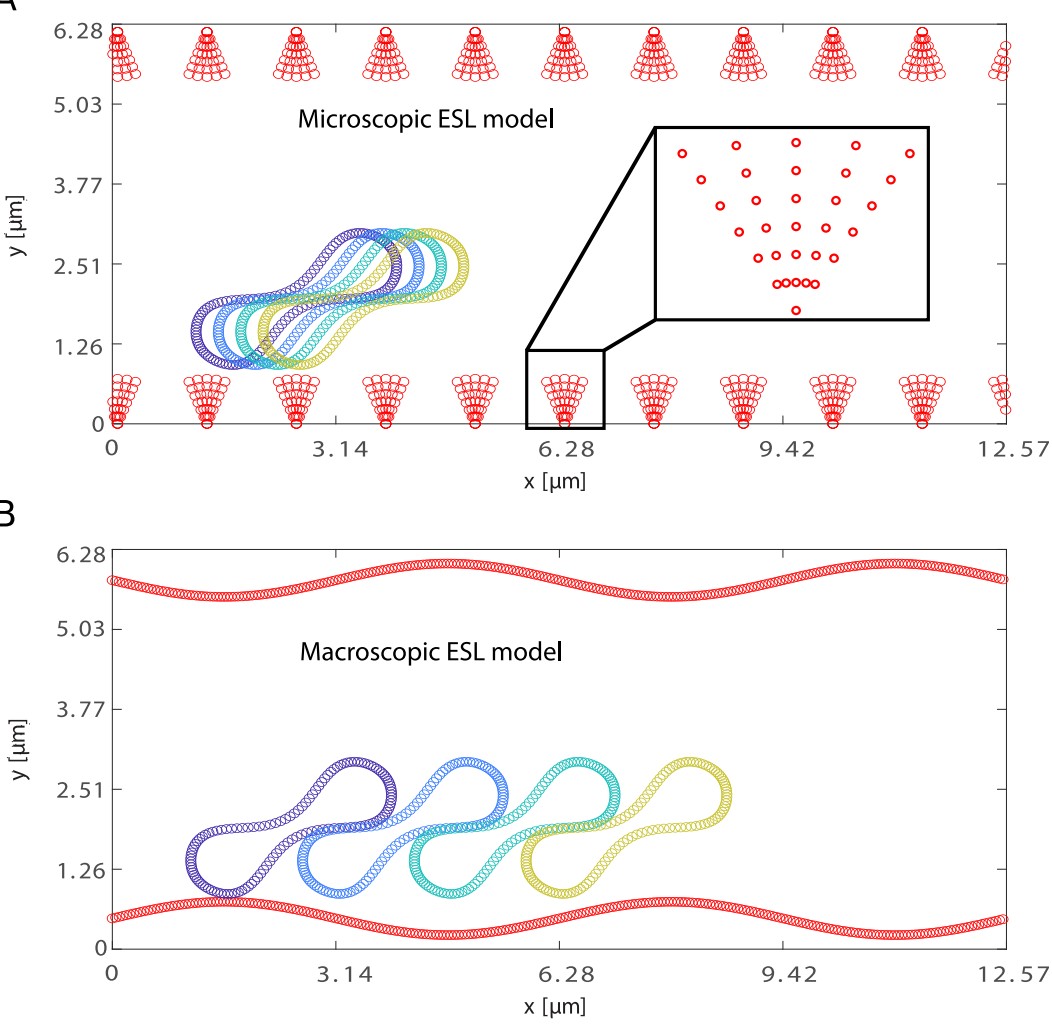

**Fig 2. Schematic of the vessel wall layout and the initial positions of the RBC.** (A) the microscopic ESL model and (B) the macroscopic ESL model. In both cases, the vessel wall layout is plotted in red.

constants, $k_{\text{tether}}$. The corresponding Lagrangian forces are

$$\boldsymbol{F}^0_{\text{ESL}} = -k_{\text{tether}}\left(\hat{\boldsymbol{X}}_0(\hat{q}, t) - \boldsymbol{Z}_0(t)\right), \tag{9}$$

$$\boldsymbol{F}^L_{\text{ESL}} = -k_{\text{tether}}\left(\hat{\boldsymbol{X}}_h(\hat{q}, t) - \boldsymbol{Z}_L(t)\right). \tag{10}$$

In the macroscopic ESL model, we coarse-grained the vessel wall into a continuous sinusoidal wave, which can be interpreted as an approximation to the variation of the ESL in the first model (see Figs 1A and 2B). In this case, all the Lagrangian points representing the wall are connected to tether points. In all cases, tether points are only implemented to Lagrangian points for the vessel wall, which correspond to the red dots in Fig 1.

## Immersed boundary method

To couple the models for blood motion, the RBC, and the vessel wall, we employed an immersed boundary method as in [35, 57]. We make use of the discretized Dirac delta functions to transfer information between the Eulerian and Lagrangian coordinates. The Eulerian force density in Eq 3 induced by the RBC and the vessel wall is determined by the Lagrangian force density via

$$f_{\text{IB}}(\boldsymbol{x}, t) = \int_q \boldsymbol{F}_{\text{IB}}(q, t)\delta(\boldsymbol{x} - \boldsymbol{X}(q, t)) \, dq, \tag{11}$$

where $\boldsymbol{F}_{\text{IB}} = \boldsymbol{F}_{\text{RBC}} + \boldsymbol{F}_{\text{ESL}}^0 + \boldsymbol{F}_{\text{ESL}}^L$.

The impermeable immersed object moves at a velocity $\boldsymbol{U}(q, t)$ equal to the local fluid velocity. This may be expressed in terms of the Dirac delta function by

$$\boldsymbol{U}_{\text{RBC}}(q, t) = \int_\Omega \boldsymbol{u}(\boldsymbol{x}, t)\delta(\boldsymbol{x} - \boldsymbol{X}(q, t)) \, d\boldsymbol{x}, \tag{12}$$

$$\boldsymbol{U}_{\text{ESL}}^{0,L}(\hat{q}, t) = \int_\Omega \boldsymbol{u}(\boldsymbol{x}, t)\delta\left(\boldsymbol{x} - \hat{\boldsymbol{X}}_{0,L}(\hat{q}, t)\right) \, d\boldsymbol{x}, \tag{13}$$

To study the effect of permeability in the Results section, Eq 13 must be adjusted to account for porosity

$$\boldsymbol{U}_{\text{ESL}}^{0,L}(\hat{q}, t) = -U_p^{0,L}\hat{n}_{0,L} + \int_\Omega \boldsymbol{u}(\boldsymbol{x}, t)\delta\left(\boldsymbol{x} - \hat{\boldsymbol{X}}_{0,L}(\hat{q}, t)\right) \, d\boldsymbol{x}, \tag{14}$$

where $\hat{n}$ is a unit normal vector to the wall and $U_p^{0,L}$ are defined as

$$U_p^0 = -k_{\text{p}}^0 \frac{\boldsymbol{F}_{\text{ESL}}^0 \cdot \hat{n}_0}{\| \hat{\boldsymbol{X}}_0 \|}, \tag{15}$$

$$U_p^L = -k_{\text{p}}^L \frac{\boldsymbol{F}_{\text{ESL}}^L \cdot \hat{n}_L}{\| \hat{\boldsymbol{X}}_L \|}. \tag{16}$$

Together, $U_p^0\hat{n}_0$ and $U_p^L\hat{n}_L$ correspond to the slip velocities given by Darcy's law for the top and bottom wall respectively [36, 58, 59]. Here we assume the porosity constants, $k_{\text{p}}^0$ and $k_{\text{p}}^L$, for the top and bottom layer are equal. As stated in [36, 58] the porosity constant $k_{\text{p}}$ is derived from Darcy's law [60]. In particular, $k_{\text{p}}$ is proportional to the number density of the pores per unit length of the layer. That is, for a fixed layer thickness increasing the permeability is equivalent to increasing the number of pores, thus increasing the blood flux through the layer.

The model is spatially discretized using a finite difference method as in [61] and temporally using a second-order accurate predictor-corrector time-stepping scheme as in [50]. In Appendix A in S1 Text, we provide a detailed description of how the spatial discretization is formulated. As a validation of our model, in Results we simulated the wall-induced migration of a single deformable membrane and compare to prior results from [31].

To study the influence of the vessel wall geometry on the motion of a single RBC near the wall, we performed simulations on a rectangular domain of size $4\pi\mu$m by $2\pi\mu$m partitioned into a Cartesian mesh of $256 \times 128$ grid points. For studying the formation of CFL using extracted vessel wall geometry, we performed simulations on a rectangular domain of size $80\mu$m by $40\mu$m partitioned into a Cartesian mesh of $1630 \times 815$ grid points, keeping the grid resolution the same as in the single RBC simulations. As shown in Appendix B in S1 Text, the

simulation results are sufficiently well-resolved with this choice of mesh size. The Lagrangian point spacing is picked to be half of the domain mesh spacing. The corresponding discretized energies of Eqs 5–7 used in simulations are given in Appendix A in S1 Text. To minimize the effect of the vessel wall topography on the drag and lift force, we calculated both forces by averaging over four initial conditions that either uniformly sample the distance between two neighboring bundles for the microscopic ESL model or one wavelength of the sine wave for the macroscopic ESL model (see Fig 2). Simulations were run till $T$ = 50s so that the center of mass of the RBC reaches a steady motion in the vertical direction for the impermeable ESL (see Figs 3A and 4A). The lift and drag were calculated as functions of time by summing the forces at each immersed boundary point at a fixed saving time step as described in [62]. For the lift, we took the opposite sign of the sum. With this approach the lift force is defined as the force acting perpendicular to the direction of flow, and the drag is defined as the force induced by the vessel wall that prevents the RBC from moving along the flow direction.

## Results

### Spatial variation of the vessel wall has a minimal effect on the motion of RBC for impermeable layers

Interactions between circulating RBCs and vessel walls are central to immune [63] and inflammatory responses [64]. Following acute injury and inflammatory conditions such as sepsis, disruption of the ESL leads to changes in the vessel wall thickness and spatial variation [43]. To study the wall-induced migration of a RBC when it is within a close proximity to the wall, we used an immersed boundary method (see Methods). The model includes two walls coupled with features of the ESL with one on the top and one at the bottom of the channel and a single RBC. The biconcave shape of the RBC is generated using the parametrization given in [65]. We explicitly include the effect of spatial variation and thickness by modeling each wall as a collection of elastic fiber bundles (Fig 5A). We assume that there are 10 bundles evenly spaced across the domain. In our model that corresponds to a healthy ESL, we include 5 fibers tethered to the root of the bundle consistent to the reported ESL structure [24]. The wall thickness, $h$, is defined to be the maximum height of a bundle. Spatial variation of the wall is given by the density of bundles and is defined to be the inverse of the distance between the roots of two neighboring bundles. In exploring the effect of density of bundles, we decreased the number of bundles until there are two bundles left, mimicking the situation occurs in sepsis and diabetes [25, 43, 55, 66].

We focus on the combined effects of spatial variation, thickness and permeability on the migration profile of the RBC and as such, we have introduced two simplifications. First, the RBC is initialized with a 30 degree orientation (see Fig 2). Second, we do not use a large elastic spring constant to model the fiber components of the ESL but instead introduce tether forces. These simplifications decrease the computational complexity of the model and are not expected to significantly alter the conclusions. In particular, using a 30 degree orientation allows us to pick a relatively small simulation termination time by minimizing the initial transition period, during which the RBC deforms and reorients before moving away from the wall. As a simple control, we repeated the simulations for a selected sets of parameter values using a circle of the same area as the RBC. We show in Appendix D in S1 Text that the results between using a circle and the 30 degree orientation of a biconcave-shaped RBC agree qualitatively. Moreover, explicitly modeling the stiffness of fibers through tether forces is a common approach to approximate solid objects in immersed boundary methods [67] that allows for a larger simulation time-step, and is not expected to significantly alter the results. In all simulations reported here, the RBC is initialized at least four grid points away from the layer. As

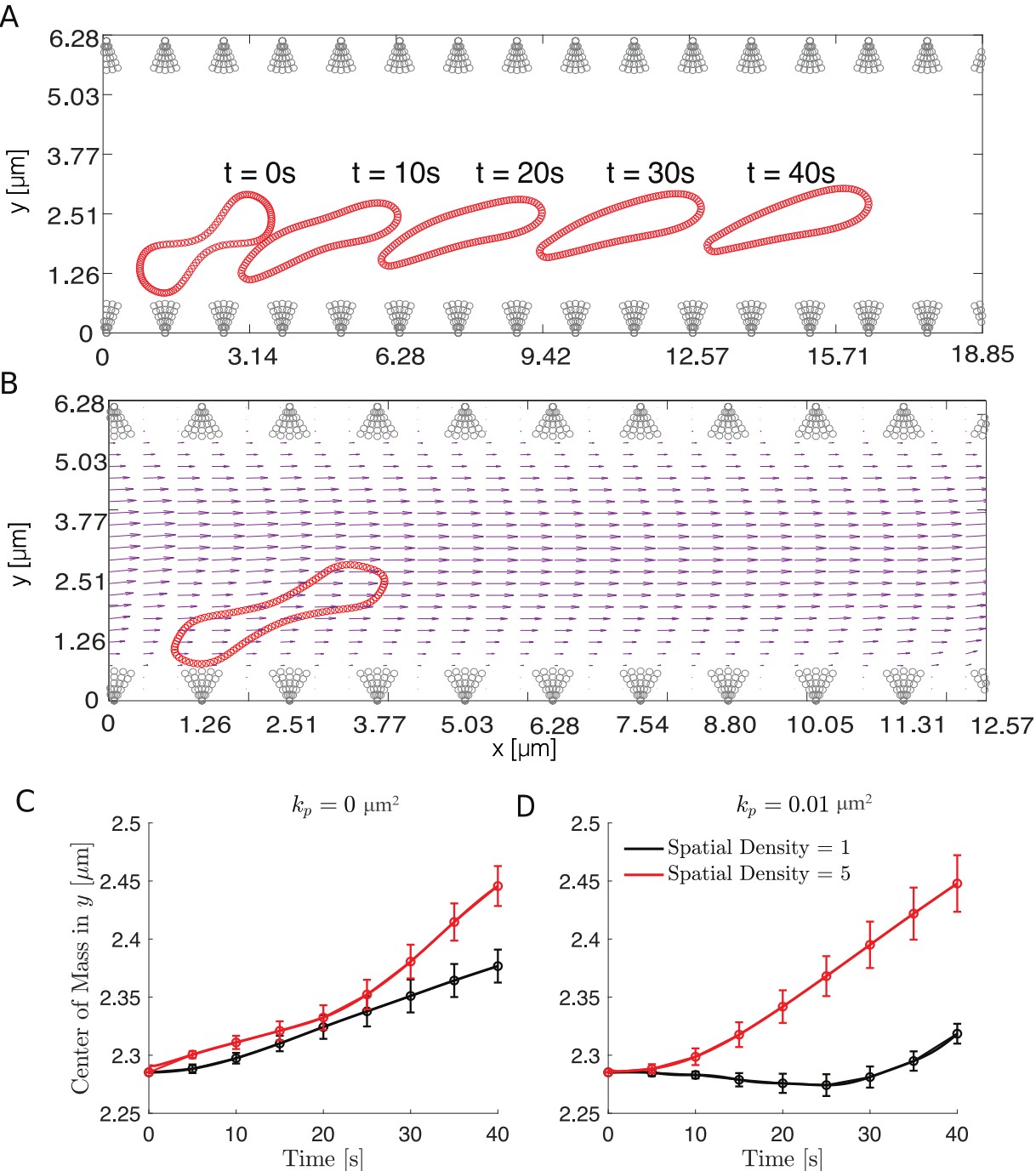

**Fig 3. The distance of the RBC's center of mass to the wall for different spatial bundle density using the microscopic ESL model.** (A) The position of the RBC is shown at indicated times measured in seconds as it moves through the channel. (B) The velocity field showing the motion of the cell on top of the blood flow at $T = 0.5$s. (C) The RBC's center of mass in the $y$ direction versus time for an impermeable wall. (D) The RBC's center of mass in the $y$ direction versus time for a highly permeable wall. In panel B and C, the black curve corresponds to a small spatial bundle density of 1 and the red curve corresponds to a large spatial bundle density of 5 with the thickness is fixed to be $h = 1.4839 \mu$m. The 95% confidence intervals are given as error bars at each data point. Parameters are summarized in Table 1.

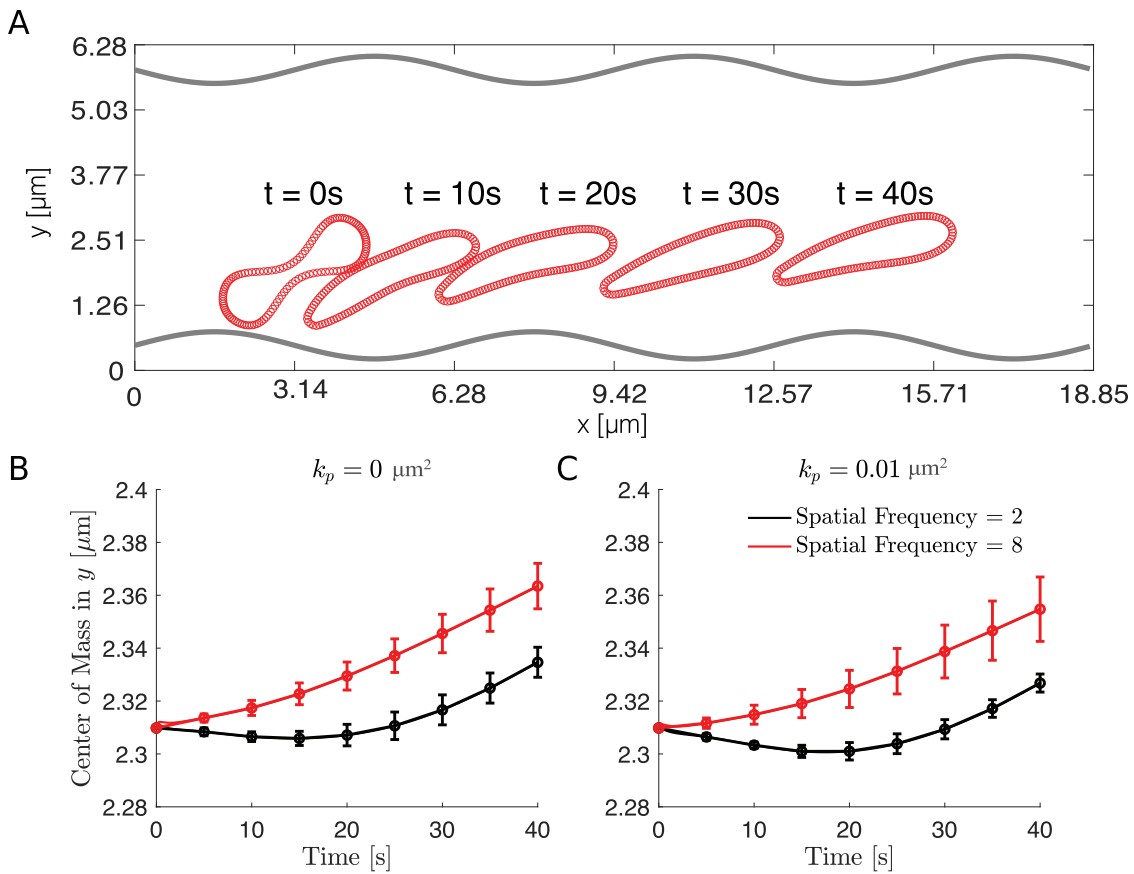

**Fig 4. The distance of the RBC's center of mass to the wall for different spatial frequency using the macroscopic ESL model.** (A) The position of the RBC is shown at indicated times as it moves through the channel. (B) The RBC's center of mass in the $y$ direction versus time for an impermeable wall. (C) The RBC's center of mass in the $y$ direction versus time for a highly permeable wall. In panel B and C, the black curve corresponds to a small spatial frequency of 2 and the red curve corresponds to a large spatial frequency of 8, and 95% confidence intervals are plotted at each data point. In all simulations, the thickness is fixed to be $h = 1.4839\mu$m. Parameters are summarized in Table 2.

such, the RBC is sufficiently far from the layer to resolve the flow using the standard immersed boundary method without lubrication corrections.

As a first output of the model, we analyzed the distance of the RBC's center of mass in the $y$ direction to the wall for an impermeable wall and a highly permeable wall. We find that the RBC moves away from the vessel wall faster when the density of bundle is large (Fig 3C and 3D). To further investigate the effect of the density of bundles as the permeability is increased, we calculated the drag and lift force of the RBC averaged over time and four initial conditions that uniformly sample the distance between two neighboring bundles (Fig 2A) as the density of bundles was increased. The absolute value of both the drag and lift are reported with 95% confidence intervals created using the four initial conditions. Interestingly, when using different densities of bundles, we find that decreasing density of bundles has a minimal effect in the drag force, with a maximum increase of 1%, and the lift force, with a maximum decrease of 14%, when the wall is impermeable or less permeable, corresponding to a healthy scenario. As the wall's permeability is increased, corresponding to the damaged/disturbed ESL, decrease in the density of bundles leads to a more apparent decrease in the lift force, a maximum decrease

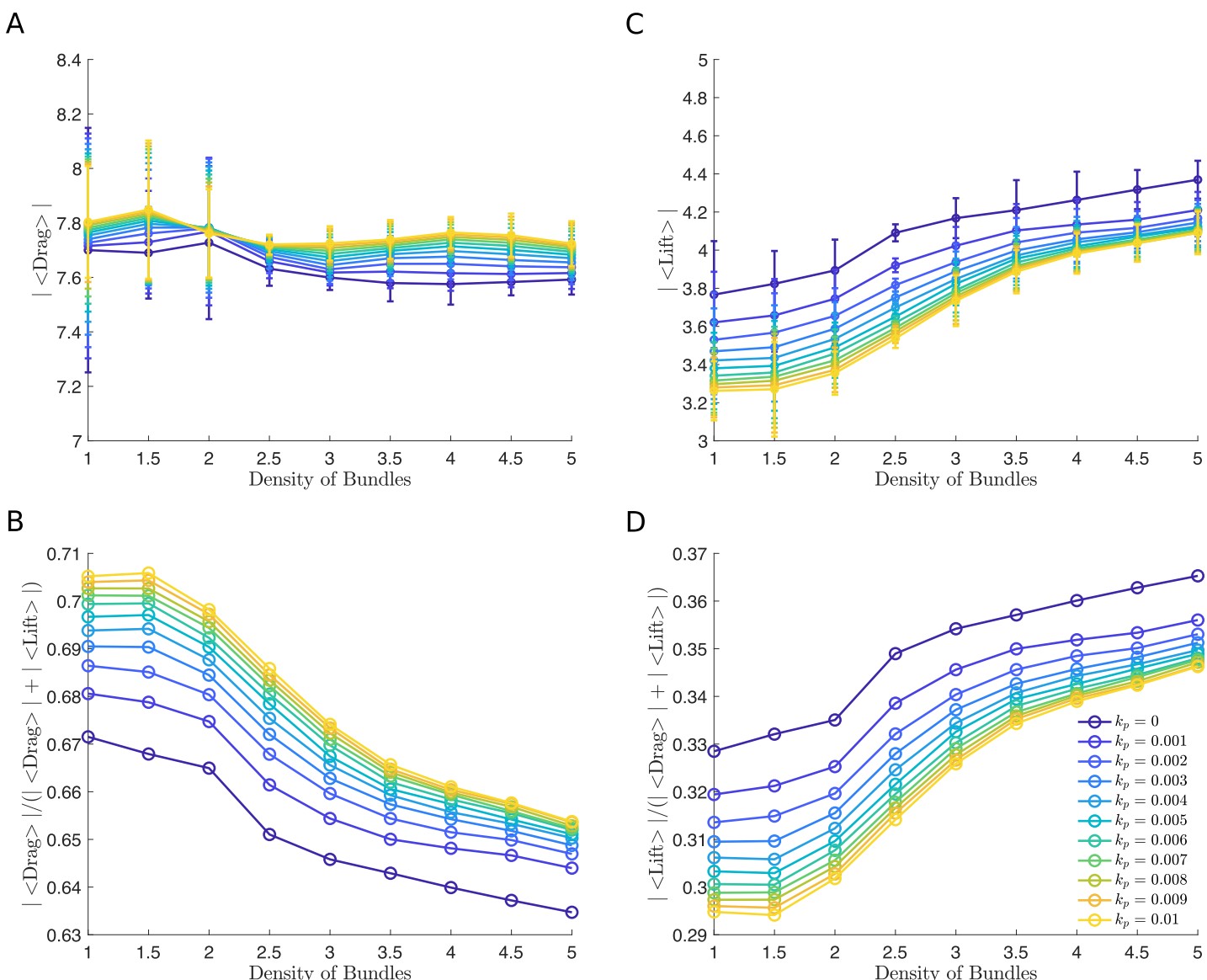

**Fig 5. The effect of changes in the density of bundles using the microscopic ESL model.** (A and C) Time and initial condition averaged drag and lift force versus density of bundles for different values of permeability. (B and D) Time and initial condition averaged fraction of drag ($|\langle\text{Drag}\rangle|/(|\langle\text{Drag}\rangle| + |\langle\text{Lift}\rangle|)$) and lift force ($|\langle\text{Lift}\rangle|/(|\langle\text{Drag}\rangle| + |\langle\text{Lift}\rangle|)$) versus density of bundles. In all simulations, the thickness is fixed to be $h = 1.4839\mu$m. In panel A and C a 95% confidence intervals are plotted at each data point. Parameters are summarized in Table 1.

of 25%, but the change in the drag force remains largely unchanged, a maximum increase of 1% (Fig 5A and 5C).

We quantified the wall-induced migration of the RBC by calculating the drag and lift over the total amount of force (Fig 5B and 5D). Plots of such fractions over the density of bundles show the effect of varying the density of bundles depends on the permeability. When the wall is highly permeable, both the drag and lift force demonstrate a more significant change (Fig 6A). This means that the RBC is more capable of moving away from a highly permeable wall when the density of bundles is increased, which is caused by a decrease in the drag (Fig 5A) and an increase in the lift (Fig 5B). On the other hand, the

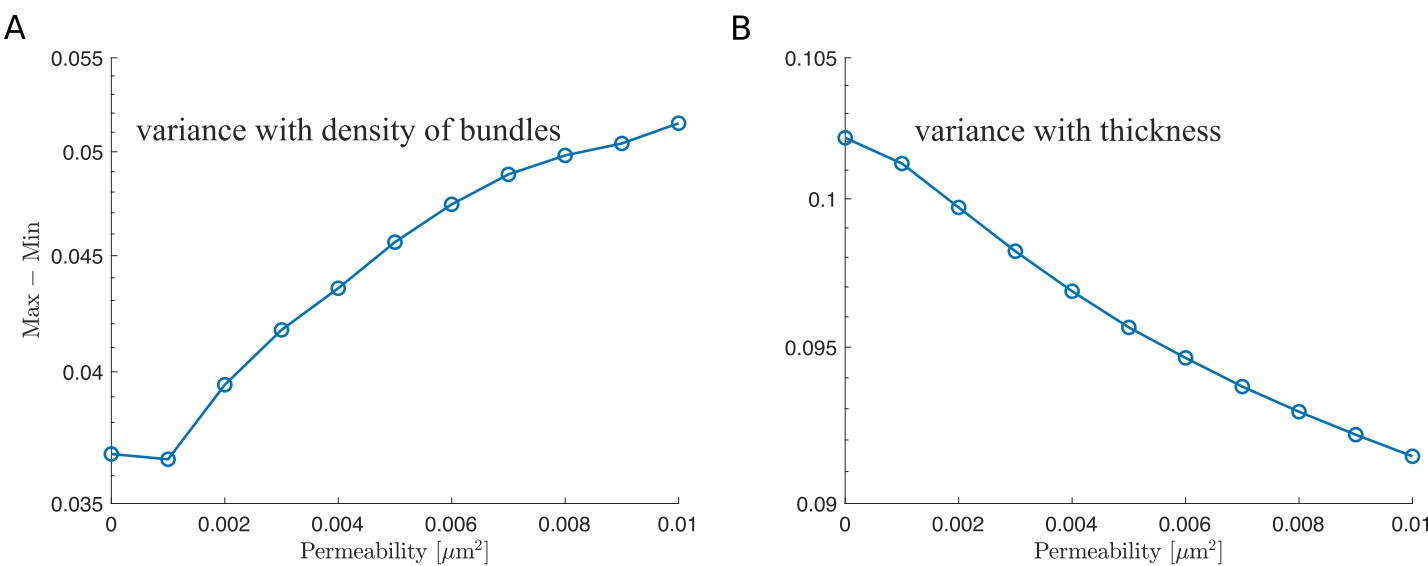

**Fig 6. The difference between the maximum and minimum values of the relative drag versus permeability using the microscopic ESL model.** (A) The difference between the maximum and minimum values of the relative drag for varying the density of bundles. Each data point is obtained using data from Fig 5B and 5D. (B) The difference between the maximum and minimum values of the relative drag for varying the thickness. Each data point is obtained using data from Fig 8B and 8D. Note that since the relative drag ($|\langle \text{Drag} \rangle|/(|\langle \text{Drag} \rangle| + |\langle \text{Lift} \rangle|)$) and relative lift ($|\langle \text{Lift} \rangle|/(|\langle \text{Drag} \rangle| + |\langle \text{Lift} \rangle|)$) sum up to 1, the variance remains the same for both the relative drag and relative lift.

motion is less affected as the density of an impermeable wall increases (Fig 5C and 5D). Taken together, we find that the spatial variation of the vessel wall plays a more dominant role in affecting the migration of the RBC when the layer is highly permeable, but minimal when the layer is less permeable.

## A thicker vessel wall can slow down the migration of the RBC

Our detailed ESL model makes it possible to embed physical features of the ESL, which are similar to those observed in experiments, in the vessel wall geometry. In addition to the spatial variation, another way in which the ESL becomes disturbed in pathological states is through the change in thickness. For instance, ESL degradation in diabetes leads to a thinner layer [43, 68] while a thicker ESL is observed during edema formation [66]. As a result, the vessel wall thickness is changed at the same time. To determine the importance of the wall thickness, we

**Table 1. Parameters for the microscopic ESL model.** Figs (5–8).

| Parameter | Description | Value |
|---|---|---|
| Re | Reynolds number | 0.01 [9, 50] |
| $k_s^{\text{RBC}}$ | Elastic spring constant | $3 \, \mu\text{N/m}$ [65] |
| $k_b$ | Bending constant | $2 \times 10^{-19} \, \text{N} \cdot \text{m}$ [65] |
| $k_a$ | Area preserving constant | $185 \, \text{N}/\mu\text{m}^2$ |
| $k_{\text{tether}}$ | Tether force constant | $3200 \, \text{N}/\mu m$ |
| $k_s^{\text{ESL}}$ | Elastic spring constant | $0.015 \, \mu\text{N/m}$ |
| $k_p$ | Porosity constant | $0 - 0.01 \, \mu\text{m}^2$ [31] |
| $h$ | Thickness of one ESL | $0.8464 - 1.7696 \, \mu\text{m}$ [55] |

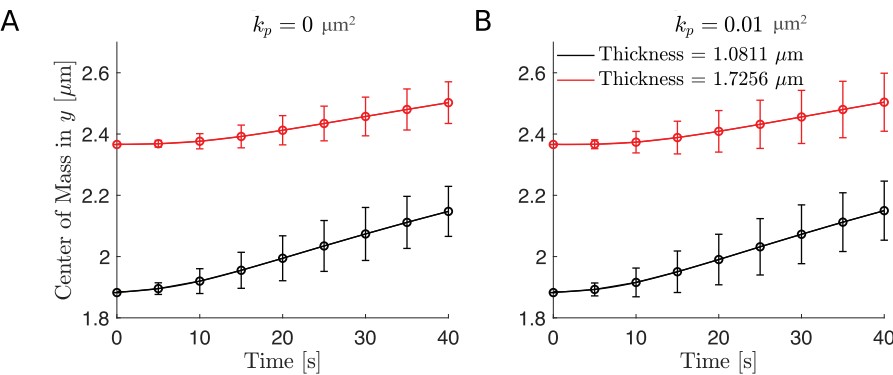

**Fig 7. The distance of the RBC's center of mass to the wall for different thickness using the microscopic ESL model.** (A) The RBC's center of mass in the $y$ direction versus time for an impermeable wall. (B) The RBC's center of mass in the $y$ direction versus time for a highly permeable wall. In both panels, the black curve corresponds to a thinner wall of thickness $1.0811\mu m$ and the red curve corresponds to a thicker wall of thickness $1.7256\mu m$. In both panels, 95% confidence intervals are plotted at each data point. In all simulations, we assume the ESL is in a healthy condition corresponding to a density of bundles of 5. Parameters are summarized in Table 1.

assume the density of bundles is fixed to 5, and increase the height of individual ESL fiber bundles, $h$, with values given in Table 1.

Examining the position of the RBC for an impermeable wall and a highly permeable wall as the thickness is increased. We find in both cases, the distance of the RBC to the wall increases more rapidly with a thinner vessel wall (Fig 7). We calculated the time and initial condition averaged drag and lift and analyzed their magnitudes. As expected, the vessel wall thickness is positively correlated to the drag and negatively correlated to the lift. For impermeable walls, increasing the thickness increased the drag force by 32% and decreased the lift force by 15%. In this case, we find that the RBC is more prone to staying near the wall due to an increasing amount of confinement. Unexpectedly, we find a similar change in the drag and lift force when the wall is highly permeable. The drag force has a maximum increase of 29% and the lift force has a maximum decrease of 14% (Fig 8A and 8C). As before, we report these results by quantifying the wall-induced migration of the RBC. We find that for all physically-relevant choices of the permeability the drag force continuously increases as the thickness is increased and consistently dominates the lift force, causing the RBC to remain near the wall due to a lack of upward lift (Fig 8B and 8D).

## The macroscopic ESL model is sufficient to capture the important features of the vessel wall associated with the ESL

The preceding microscopic ESL model demonstrates a clear relation in how the drag and lift of a RBC depend on the vessel wall thickness as permeability varies. It also suggests that the effect of the spatial variation of the wall is minimal when the wall is impermeable. To develop a more computational efficient model that captures these important physical features of the vessel wall associated with changes in the ESL, we exploited the fact that the RBC only interacts with the surface/outer region of the ESL to coarse-grain its structure into a continuous sine wave of the form $y(x) = A\sin(ax) + h$ where $(A + h)$ represents the thickness of a *single* wall either on the top or at the bottom and the spatial variation of the wall is characterized by the spatial frequency, which is given by the frequency $a$ of the sine wave. Throughout this work we fix the sinusoidal amplitude to be $A = 0.336\mu m$. To demonstrate the difference in the computational cost between the coarse-grained and the microscopic ESL model, we simulated a dense

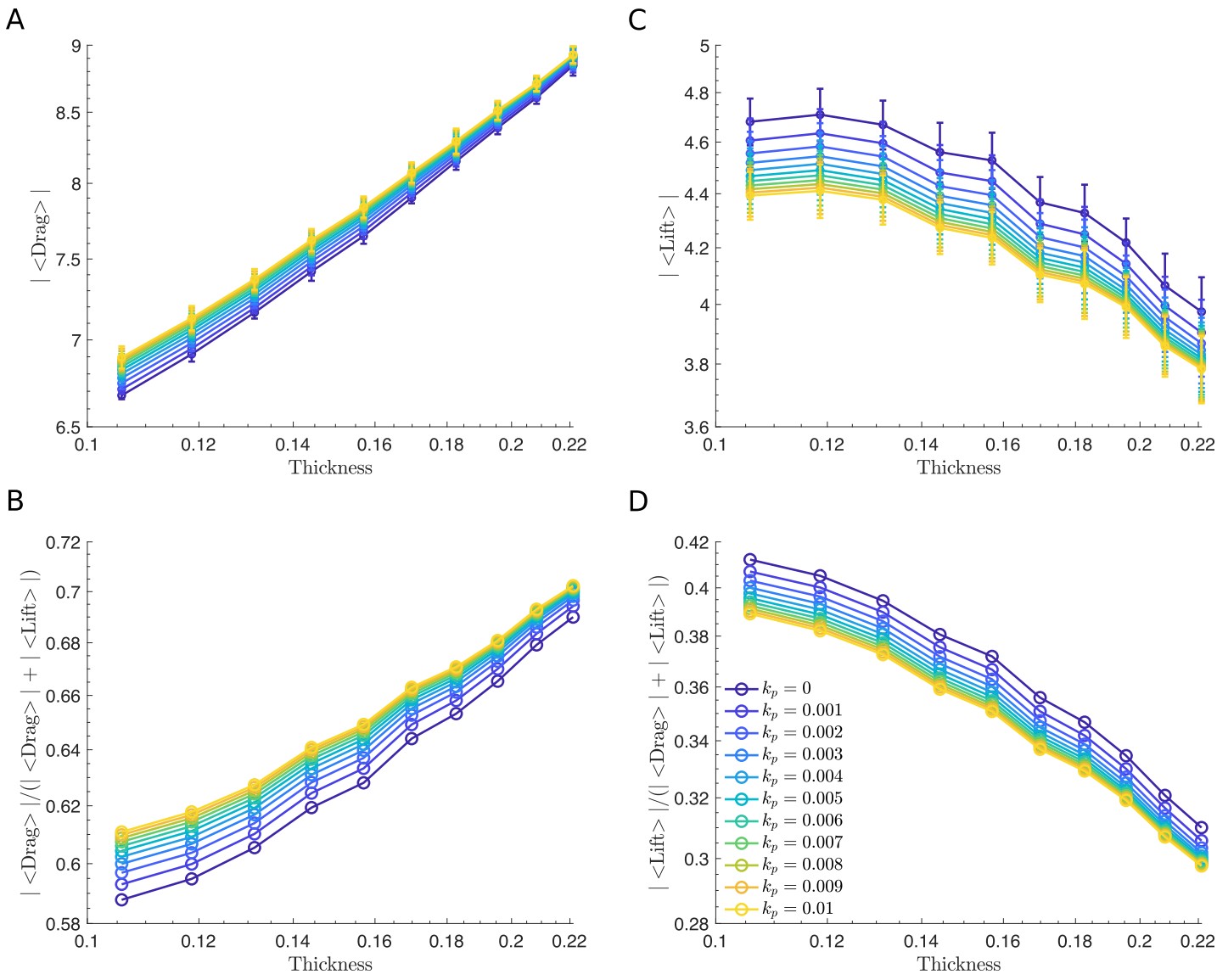

**Fig 8. The effect of variation in ESL thickness using the microscopic ESL model.** (A and C) Time and initial condition averaged drag and lift force versus wall thickness for different values of permeability. (B and D) Time and initial condition averaged fraction of drag ($|\langle \text{Drag}\rangle|/(|\langle \text{Drag}\rangle| + |\langle \text{Lift}\rangle|)$) and lift force ($|\langle \text{Lift}\rangle|/(|\langle \text{Drag}\rangle| + |\langle \text{Lift}\rangle|)$) versus thickness over all values of permeability. In all simulations, we assume the ESL is in a healthy condition corresponding to a density of bundles of 5. In panel A and C a 95% confidence intervals are plotted at each data point. Parameters are summarized in Table 1.

case of the ESL using both models with thickness fixed for $k_{\text{p}} = 0$, 0.005, and 0.01$\mu$m$^2$. As shown in Appendix E in S1 Text, the macroscopic ESL model is approximately twice as fast as the microscopic ESL model and the speed-up factor increases for larger values of $k_{\text{p}}$.

As before, we first examined the position of the RBC to the wall (Figs 4 and 9) and calculated the drag and lift averaged over the course of the simulation for four initial conditions that uniformly sample one wavelength of the sine wave (Fig 2B). Using the same set of permeability values, we then calculated the resulting drag and lift. We find that increasing the spatial frequency only decreases the wall-induced migration of the RBC when the wall is highly permeable but does not affect its migration when the wall is nearly impermeable (Figs 10 and 11). When exploring the influence of the wall thickness, we find that the migration of the RBC is

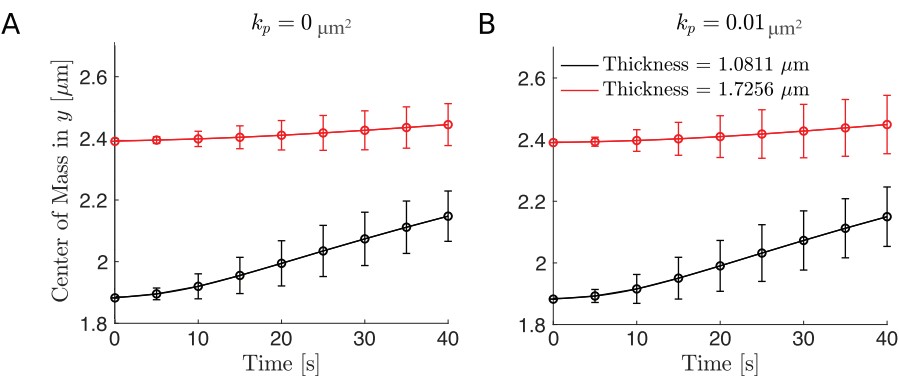

**Fig 9. The distance of the RBC's center of mass to the wall for different thickness using the macroscopic ESL model.** (A) The RBC's center of mass in the $y$ direction versus time for an impermeable wall. (B) The RBC's center of mass in the $y$ direction versus time for a highly permeable wall. In both panels, the black curve corresponds to a thinner wall of thickness $1.0811\mu m$ and the red curve corresponds to a thicker wall of thickness $1.7256\mu m$. In both panels 95% confidence intervals are plotted at each data point. In all simulations, we assume the ESL is in a healthy condition corresponding to a spatial frequency of 2. Parameters are summarized in Table 2.

significantly inhibited when a thick wall is present and such a dependency persists for all values of permeability (Figs 11 and 12).

## Cell-free layer estimation for healthy and disrupted ESL

Under different pathological conditions, changes in the ESL can lead to drastic changes in the vessel wall geometry. For instance, under infection or septic conditions, damage of the ESL occurs, enabling leukocyte and platelet adhesion. As a result, the vessel wall thickens and becomes bumpier, disrupting the microcirculation [43, 66]. To understand how the microcirculation is changed in a more realistic setting, we focused on analyzing the CFL thickness and mean RBC velocity estimated from simulations using the macroscopic ESL model (see Materials and methods). The vessel wall geometries are directly extracted from medical images taken using intravital microscopy provided in [43] (see Fig 13A and 13D). The vessel wall dimensions are measured by calculating the distance between the erythrocytes and the endothelium using the intravital microscopic images [43]. To extract the vessel wall geometries we used [69]. In the septic case, the vessel geometry is affected by the leukocytes that are sticking to the ESL (Fig 13D). Therefore, we manually trace the boundary of the blood vessel, including the adhered leukocytes, to extract the vessel wall geometry. To systematically study the formation

**Table 2. Parameters for the macroscopic ESL model.** (Figs 10–12).

| Parameter | Description | Value |
|---|---|---|
| Re | Reynolds number | 0.01 [9, 50] |
| $k_s^{RBC}$ | Elastic spring constant | 3 $\mu N/m$ [65] |
| $k_b$ | Bending constant | $2 \times 10^{-19}$ N · m [65] |
| $k_a$ | Area preserving constant | 185 N/$\mu m^2$ |
| $k_{tether}$ | Tether force constant | 3200 N/$\mu m$ |
| $k_s^{ESL}$ | Elastic spring constant | 0.015 $\mu N/m$ |
| $k_p$ | Porosity constant | $0 - 0.01$ $\mu m^2$ [31] |
| $(A + h)$ | Thickness of one ESL | $0.8464 - 1.7696$ $\mu m$ [55] |
| $a/(2\pi)$ | Spatial frequency | $1 - 10$ |

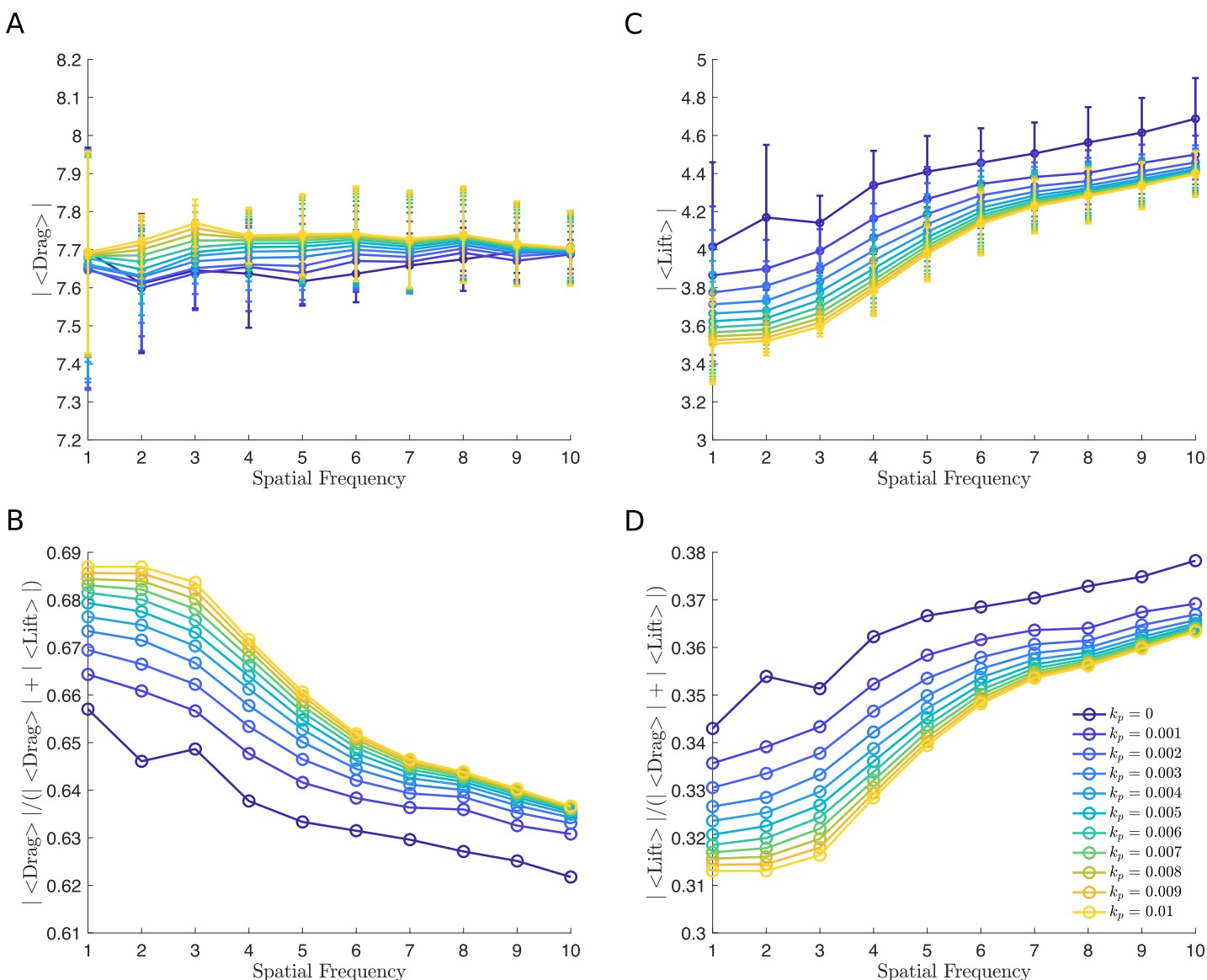

**Fig 10. The macroscopic ESL model illustrates a similar qualitative effect of spatial variation.** (A and C) Time and initial condition averaged drag and lift force versus layer spatial frequency. (B and D) Time and initial condition averaged fraction of drag ($|\langle \text{Drag}\rangle|/(|\langle \text{Drag}\rangle| + |\langle \text{Lift}\rangle|)$) and lift force ($|\langle \text{Lift}\rangle|/(|\langle \text{Drag}\rangle| + |\langle \text{Lift}\rangle|)$) versus spatial frequency over all values of permeability. In all simulations, the thickness is fixed to be $(A + h) = 1.4839 \mu$m. In panel A and C 95% confidence intervals are plotted at each data point. Parameters are summarized in Table 2.

of CFL using experimental data, we first consider the scenario where the hematocrit is fixed. The number of RBCs in the healthy condition is estimated to be 48. In the disrupted septic condition, the number of RBCs is chosen so that the cell density is the same as in the healthy case. We assume that in both cases RBCs are placed uniformly across the region bounded by the healthy ESL initially.

We begin by considering a system that contains only RBCs using parameters summarized in Table 3. To measure the CFL thickness, we averaged 800 vertical slices of the steady-state plot showing cells' positions (see Fig 13 and S1 and S2 Videos) evenly spaced from $x = 10 \mu$m to $x = 70 \mu$m. In each slice, we measured the distance between the outer edge of the RBC core

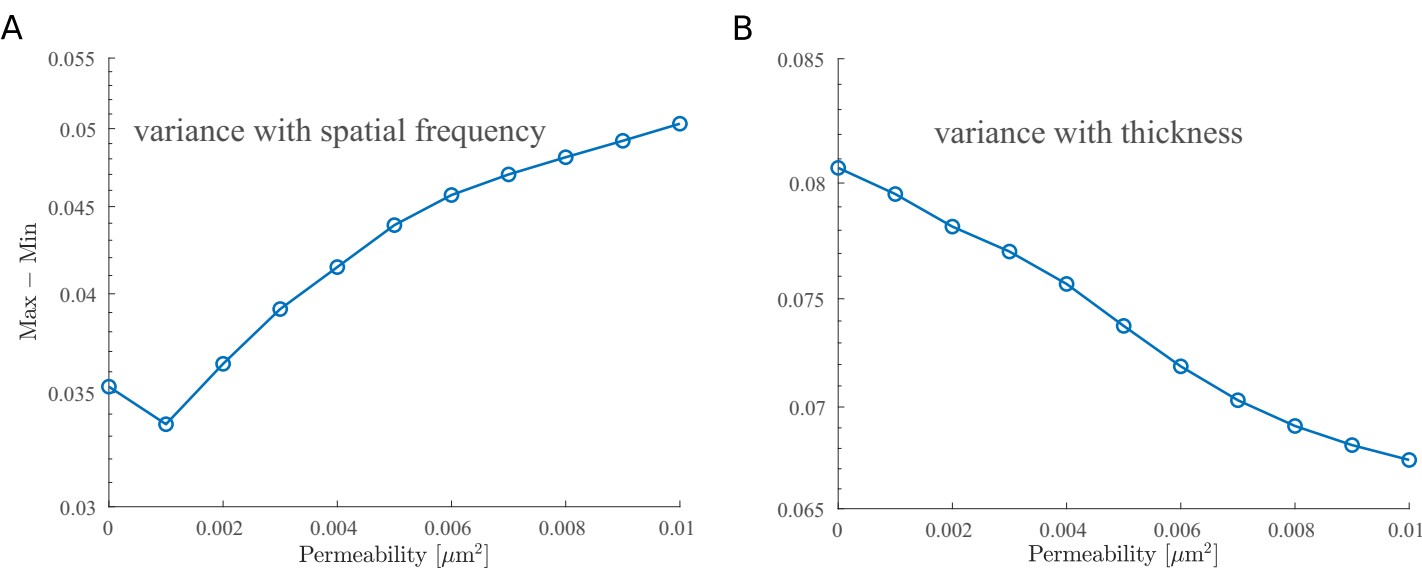

**Fig 11. The difference between the maximum and minimum values of the relative drag versus permeability using the macroscopic ESL model.** (A) The difference between the maximum and minimum values of the relative drag for varying the spatial frequency. Each data point is obtained using data from Fig 10B and 10D. (B) The difference between the maximum and minimum values of the relative drag for varying the thickness. Each data point is obtained using data from Fig 12B and 12D. Note that since the relative drag ($|\langle\text{Drag}\rangle|/(|\langle\text{Drag}\rangle| + |\langle\text{Lift}\rangle|)$) and relative lift ($|\langle\text{Lift}\rangle|/(|\langle\text{Drag}\rangle| + |\langle\text{Lift}\rangle|)$) sum up to 1, the variance remains the same for both the relative drag and relative lift.

and the ESL, similar to the CFL measurements done in experiments [18, 19]. The width of the estimated CFL is then obtained by summing up the distance over all slices and averaging over the total number of slices. Comparing results from simulating the two cases, we find that the CFL is thinner in the septic scenario and the mean RBC velocity is slower (see the first and second row in the RBC-only category in Table 4), in agreement with those reported in [18–20] qualitatively. Further comparing the top and bottom CFL, we observe a nonaxisymmetric CFL profile with the top CFL thicker than the bottom CFL.

The plasma is a mixture of RBCs, leukocytes, and platelets. To study the formation of CFL in a more realistic set up, we generalized the model to include leukocytes. Here leukocytes are modeled as circles of radius 6 $\mu$m [72, 73]. In the healthy case, two leukocytes are randomly placed within the blood vessel. The number of leukocytes are chosen so that there is at least one in the disrupted case and the relative ratio between leukocytes and RBCs remains close to that reported from experiments [74]. Although here we assume the same leukocyte density in the healthy and septic scenarios for purposes of comparison, the leukocyte density is known to vary through the microvasculature [75]. As before, we estimated the thickness of the CFL at steady state and the mean RBC velocity in both the healthy and disrupted septic cases (see Fig 14 and S4 and S5 Videos). Examining the estimated CFL thickness and the mean RBC velocity, we again observe a decrease in both quantities in the disrupted septic case but the amount of decrease in both quantities is less than in the RBC-only model (see the first and second row in the whole blood category in Table 4). Our results are consistent with previous studies qualitatively [18–20]. As in the RBC-only model, we see a nonaxisymmetric CFL profile with the top CFL notably thicker than the bottom CFL.

Examining the image corresponds to the septic condition, we see that the blood vessel diameter is significantly reduced (Fig 13D), which could lead to a decrease in the density of cells [76, 77]. Therefore, we repeated the estimation of CFL thickness and mean RBC velocity of the septic case using a lower hematocrit for both the RBC-only and whole blood models. To

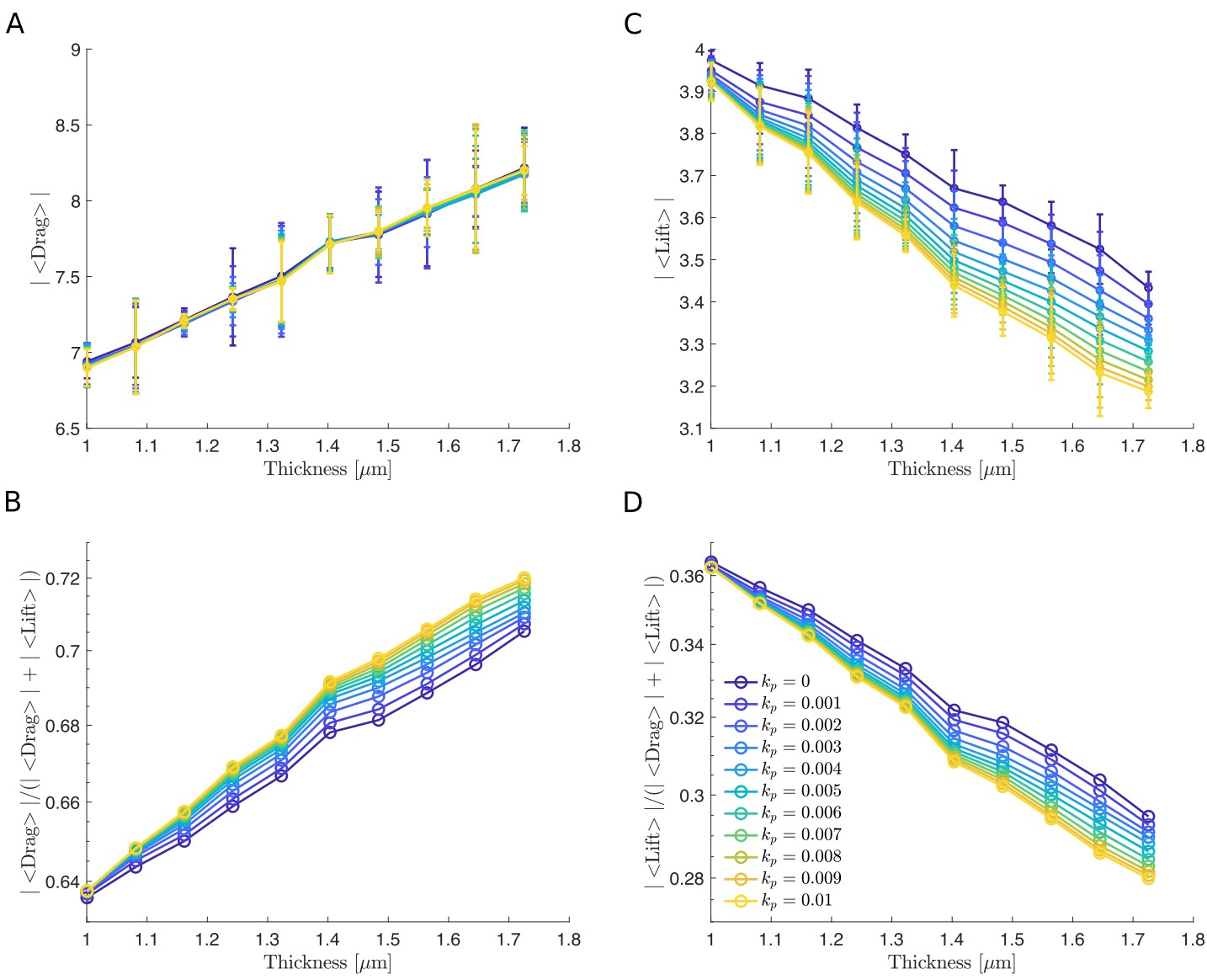

**Fig 12. The macroscopic ESL model illustrates a similar qualitative effect of thickness.** (A and C) Time and initial condition averaged drag and lift force versus wall thickness for different values of permeability. (B and D) Time and initial condition averaged fraction of drag ($|\langle Drag \rangle|/(|\langle Drag \rangle| + |\langle Lift \rangle|)$) and lift force ($|\langle Lift \rangle|/(|\langle Drag \rangle| + |\langle Lift \rangle|)$) versus thickness over all values of permeability. In all simulations, we assume the ESL is in a healthy condition corresponding to a spatial frequency of 1. In panel A and C a 95% confidence intervals are plotted at each data point. Parameters are summarized in Table 2.

determine the number of RBCs and leukocytes and their initial positions, we used the same initial set of cells as in the healthy condition and removed those ones that lie outside the region enclosed by the vessel wall. Comparing the results of using a lower hematocrit to the ones with a higher hematocrit, we find that decrease in the hematocrit leads to an increase in the CFL thickness and an increase in the mean RBC velocity in both the RBC-only and whole blood models (see Figs 15 and 16 and Table 4 and S3 and S6 Videos). For a fixed vessel diameter, we find that our results agree qualitatively with those reported in [18–20]. The nonaxisymmetric profile of the CFL is preserved in the low hematocrit case. Interestingly, the increase in the CFL thickness using a lower hematocrit is more notable in the RBC-only model than in the whole blood model. This highlights a potential role of leukocytes in affecting CFL formation.

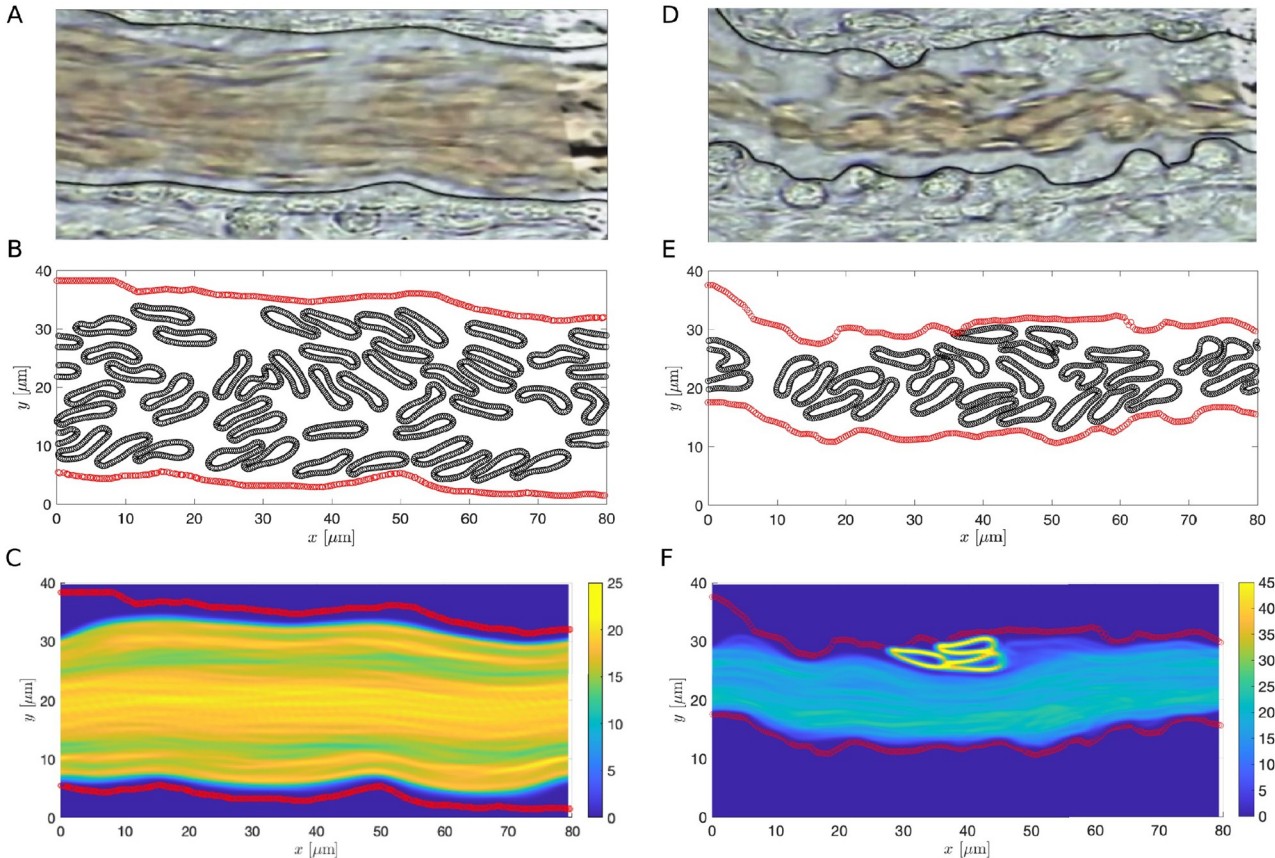

**Fig 13. RBC-only simulations with extracted vessel wall geometry using the same cell density in both cases.** (A and D) Medical images from [43] taken (A) under a healthy condition and (D) during sepsis. In both cases black curves represent the extracted geometry used in simulations. (B and E) Schematics of the ESL layout (in red) and the positions of the RBCs at $T = 200s$ with healthy (B) and disrupted ESL (E). (C and F) The steady-state density of RBC distribution. Each ESL is assumed impermeable ($k_p = 0\mu m^2$). RBC and ESL parameters are summarized in Table 3.

The thickness of the CFL is largely influenced by the vessel diameter. The fraction of the vessel diameter occupied by the CFL in arterioles in the rat cremaster muscle was found to increase from 14% to 17% for vessel diameter ranging from 5 to 8 $\mu$m but to gradually decrease from 18% to 12% as the diameter increases from 8 to 25 $\mu$m [78]. As a first comparison, we followed the procedure in [78] to study the relation between the fraction of mean CFL thickness over vessel diameter using the healthy and disrupted cases. The mean vessel diameter in the

**Table 3. Parameters for the RBC-only and the whole blood simulations.** (Figs 13 and 16).

| Parameter | Description | Value |
|---|---|---|
| Re | Reynolds number | 0.01 [9, 50] |
| $k_s^{RBC}$ | Elastic spring constant | 3 $\mu$N/m [65] |
| $k_s^{Leuko}$ | Elastic spring constant | 30 $\mu$N/m [70] |
| $k_b^{RBC}$ | Bending constant | $2 \times 10^{-19}$ N $\cdot$ m [65] |
| $k_b^{Leuko}$ | Bending constant | $2 \times 10^{-18}$ N $\cdot$ m [71] |
| $k_a$ | Area preserving constant | 185 N/$\mu m^2$ |
| $k_{tether}$ | Tether force constant | 3200 N/$\mu m$ |
| $k_p$ | Porosity constant | 0 $\mu m^2$ |

**Table 4. Steady state RBC velocity and CFL thickness.** RBC velocity in the horizontal direction and the estimated CFL thickness with extracted vessel wall geometry for both the RBC-only simulations and the whole blood simulations.

| | | Mean RBC Velocity [mm/s] | Top CFL [$\mu$m] | Bottom CFL [$\mu$m] |
|---|---|---|---|---|
| RBC-only | Healthy | 0.528 | 3.316 | 1.101 |
| | Disrupted | 0.223 | 2.29 | 1.013 |
| | Disrupted (Low Density) | 0.324 | 3.99 | 3.633 |
| Whole Blood | Healthy | 0.444 | 3.705 | 1.1 |
| | Disrupted | 0.221 | 3.63 | 1.084 |
| | Disrupted (Low Density) | 0.258 | 3.289 | 1.896 |

healthy case (Fig 13A) is estimated to be 31$\mu$m whereas in the disrupted septic case (Fig 13D) it is estimated to be 20$\mu$m. Using the RBC-only model, the fraction is estimated to be 7.1% in the healthy case. In the disrupted case the fraction increases to approximately 7.3% with the same cell density as in the healthy case and to 18% with a lower cell density. Repeating the same calculation for the whole blood model, we again find that the fraction increases as the vessel diameter is decreased. In the healthy condition where the vessel diameter is larger, the

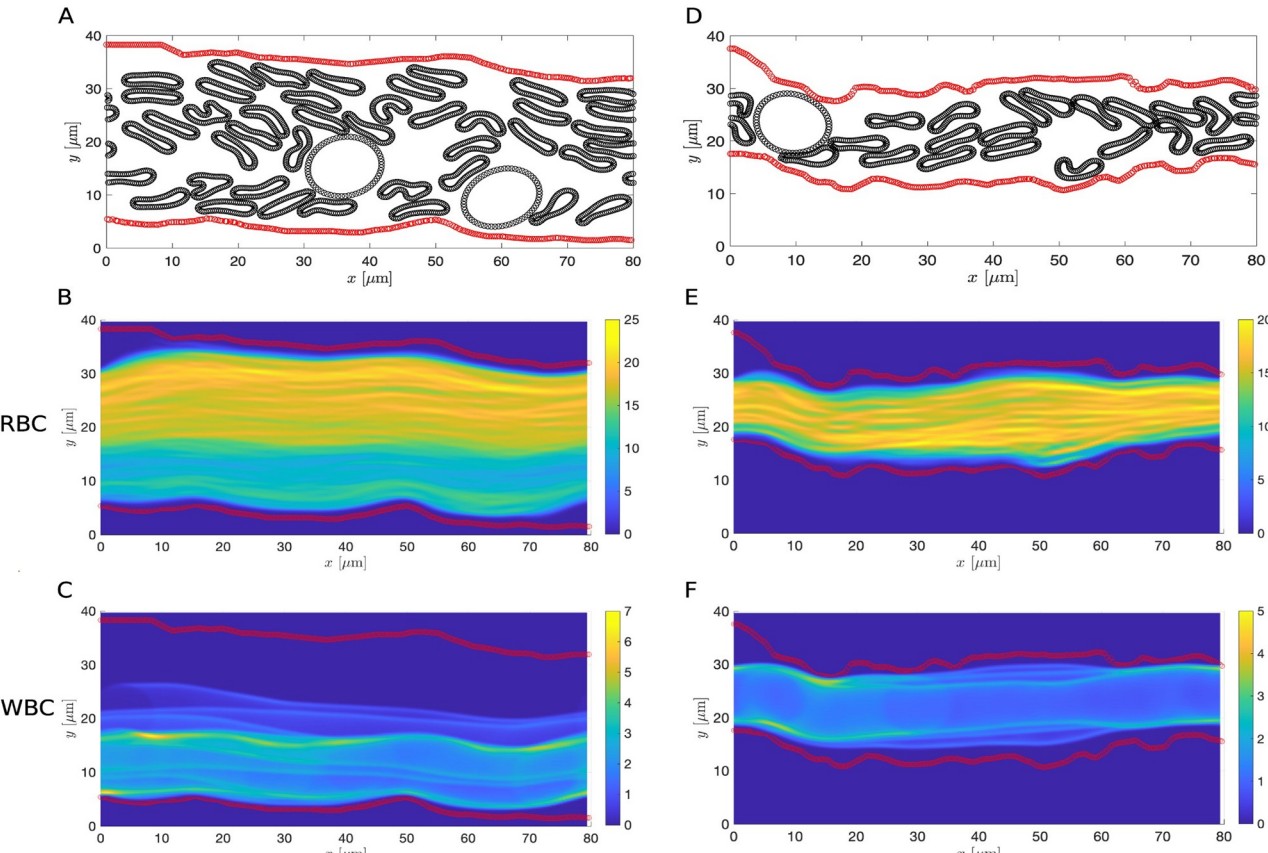

**Fig 14. Whole blood simulations with extracted vessel wall geometry using the same cell density in both cases.** (A and D) Schematics of the vessel wall layout (in red) and positions of the RBCs and leukocytes at $T = 200$s with healthy (A) and disrupted ESL (D). (B and E) The steady-state density of RBC distribution. (C and F) The steady-state density of leukocyte (WBC) distribution. Each ESL is assumed impermeable ($k_p = 0\mu$m$^2$). RBC, leukocytes and ESL parameters are summarized in Table 3.

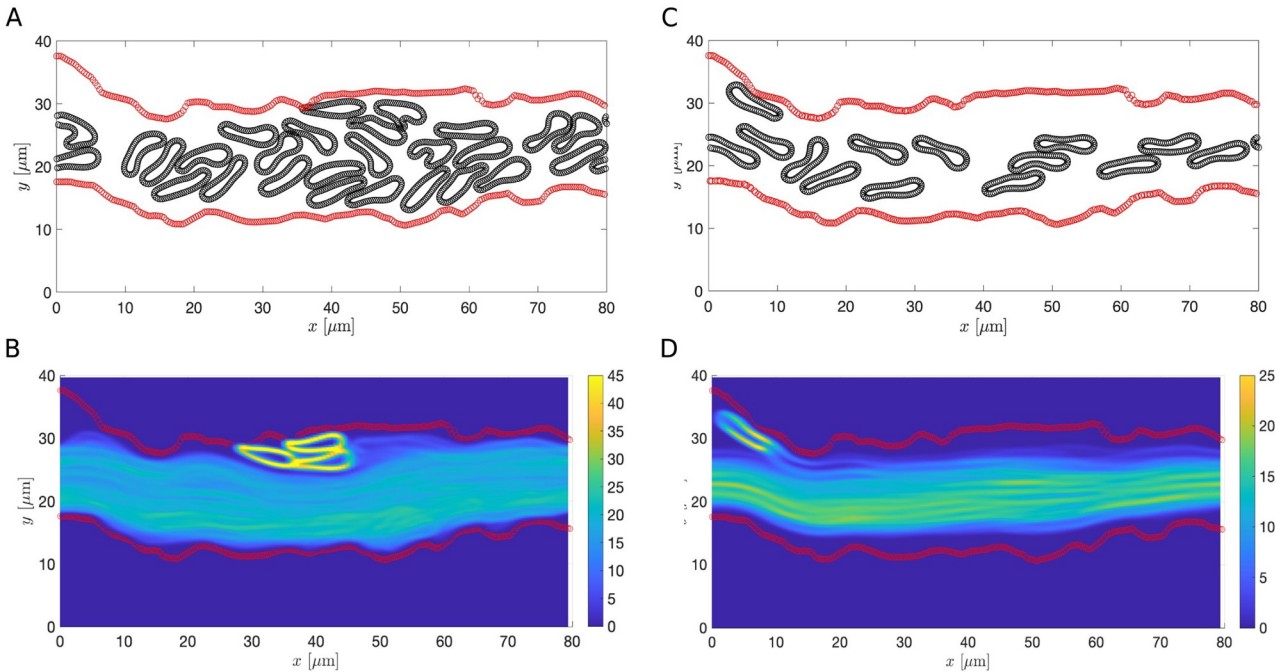

**Fig 15. Comparison of the RBC-only simulations in the septic case using different cell densities.** (A and C) Schematics of the ESL layout (in red) and the positions of the RBCs at $T = 200$s with the cell density matching the one in the healthy case (A) and a lower cell density (C). (B and D) The steady-state density of RBC distribution. Each ESL is assumed impermeable ($k_\mathrm{p} = 0\mu m^2$). RBC and ESL parameters are summarized in Table 3.

fraction is approximately 8%. In the disrupted condition, the fraction is increased to approximately 12% using the same cell density as in the healthy case and to 13% using a lower cell density. We see that regardless of the cell density the fraction increases as the mean vessel diameter is decreased in both the RBC-only and whole blood models. The overall trend we observe here agrees with [78] quantitatively.

One notable difference between Fig 13A and 13D is the overall diameter of the blood vessel. The vessel diameter is significantly decreased during sepsis. To further investigate the changes in the CFL thickness in relation to the vessel diameter, we re-simulated the RBC-only model using straight channels of width, corresponds to the mean vessel diameter in the healthy and sepsis conditions. For a fixed cell density, the estimated CFL thickness is 2.011$\mu$m in a wider channel and it decreases slightly to 1.61$\mu$m in the narrower channel (Fig 17A and 17B). Comparing the results for a fixed vessel diameter, we see that the CFL thickness increases significantly to 3.972$\mu$m when a lower cell density is used (Fig 17B and 17C). The correlation among CFL thickness, vessel diameter, and cell density is consistent with our results of using the extracted vessel wall geometry. However, we note that the spatial profile of the CFL is much less uniform across the whole vessel using experimental data than using a straight channel (Fig 13F). Taken together, the CFL formation is due to a combined effect of vessel diameter, cell density and the spatial variation of the vessel wall.

We also compared the dimensionless thickness of the CFL, defined as the fraction of the CFL thickness over the vessel diameter, $R$, to modeling results given by [79] and experimental data provided by [12] extracted using [80]. Here we used the simulated results with a lower hematocrit to perform the comparison as they give a better visual agreement to the medical image in the disrupted septic case (see Figs 13A, 13F, 14E, 15D and 16E). In Fig 18 we showed

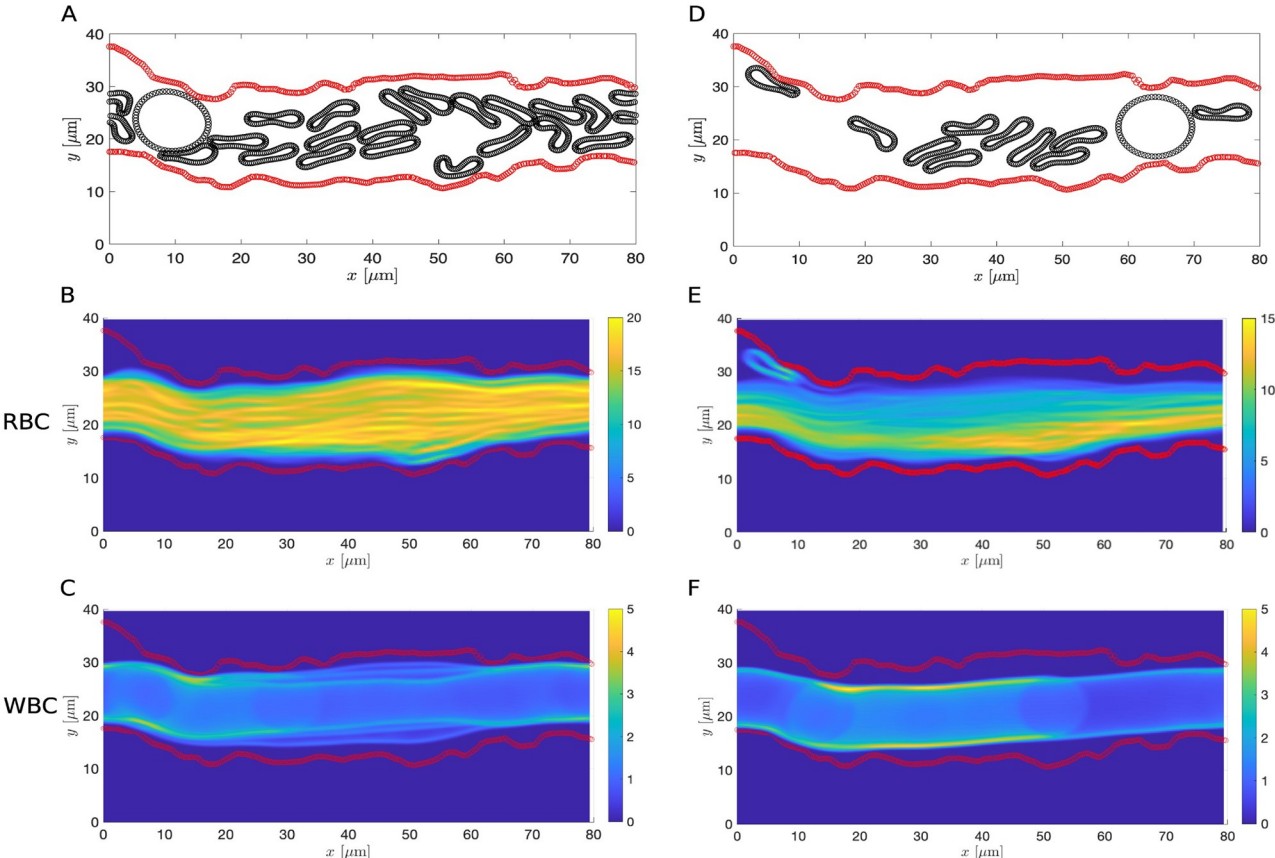

**Fig 16. Comparison of the whole blood simulations in the septic case using different cell densities.** (A and D) Schematics of the ESL layout (in red) and positions of the RBCs and leukocytes at $T = 200$s with the cell density matching the one in the healthy case (A) and a lower cell density (D). (B and E) The steady-state density of RBC distribution. (C and F) The steady-state density of leukocyte (WBC) distribution. Each ESL is assumed impermeable ($k_p = 0\mu m^2$). RBC, leukocytes and ESL parameters are summarized in Table 3.

the results of using our models against those from [12, 79] for discharge hematocrit level of 45% and 10%, corresponding to the healthy and disrupted case in our setup respectively. We see that the dimensionless CFL thickness estimated from the simulations using both the RBC-only and the whole blood models is in agreement with the available observations to statistical error. Although the extracted vessel geometry includes the attached leukocytes as the scale makes it impractical to isolate only the ESL, this provides insights into how a disrupted vessel wall geometry may alter blood flow at smaller scales.

## Discussion

The vessel wall geometry plays a critical role in regulating the circulatory system. In particular, changes in the endothelial surface layer are known to affect the wall geometry under different pathological conditions. In a healthy blood vessel, the ESL acts as an antithrombotic and anti-inflammatory agent that prevents cells from sticking to the layer, resulting in a relatively smooth vessel wall geometry. Disturbances in the ESL, such as the changes that occur during sepsis, can drastically change the vessel wall geometry and lead to changes in the blood flow and the dynamics of cells including the wall-induced migration. Using a combination of detailed and coarse-grained computational models via the immersed boundary method, we

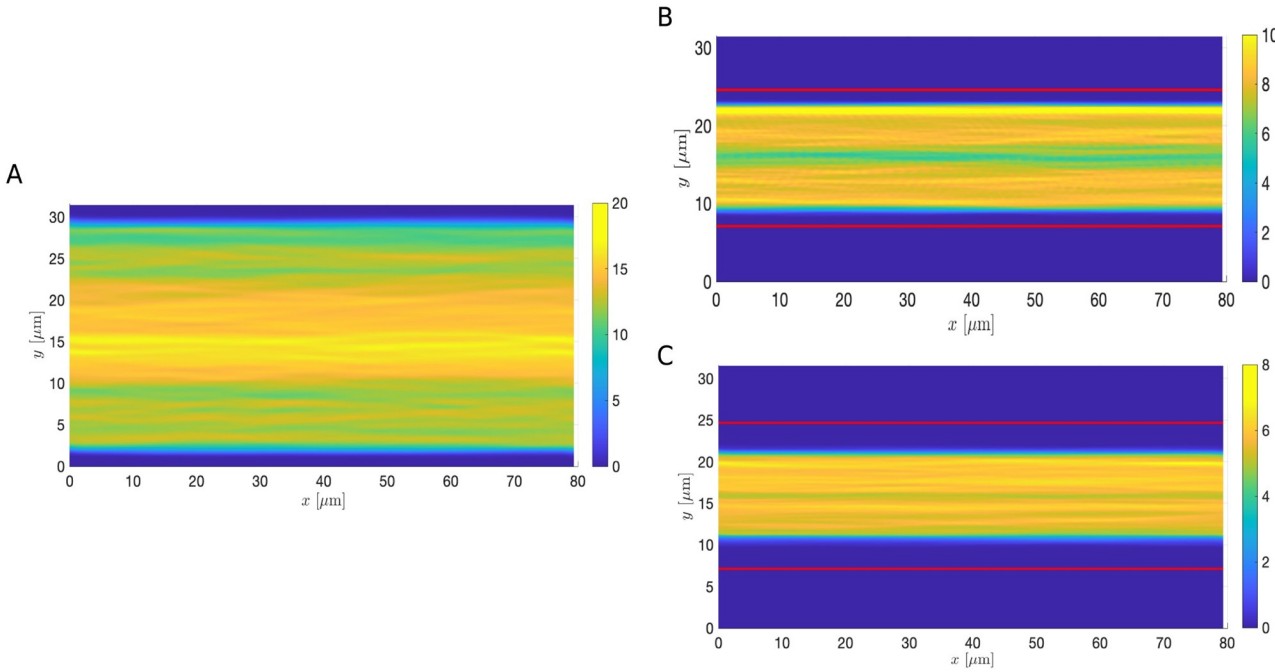

**Fig 17. RBC-only simulations in straight channels with different channel widths and RBC densities.** (A) The steady-state density of RBC distribution in a wider channel with a width of $31\mu$m. (B) The steady-state density of RBC distribution in a narrower channel with a width of $20\mu$m using the same RBC density as in (A). (C) The steady-state density of RBC distribution in a narrower channel with a width of $20\mu$m using a lower RBC density than in (A). In panel B and C the channel boundaries are plotted in solid red lines. Each ESL is assumed impermeable ($k_\mathrm{p} = 0\mu$m$^2$). RBC parameters are summarized in Table 3.

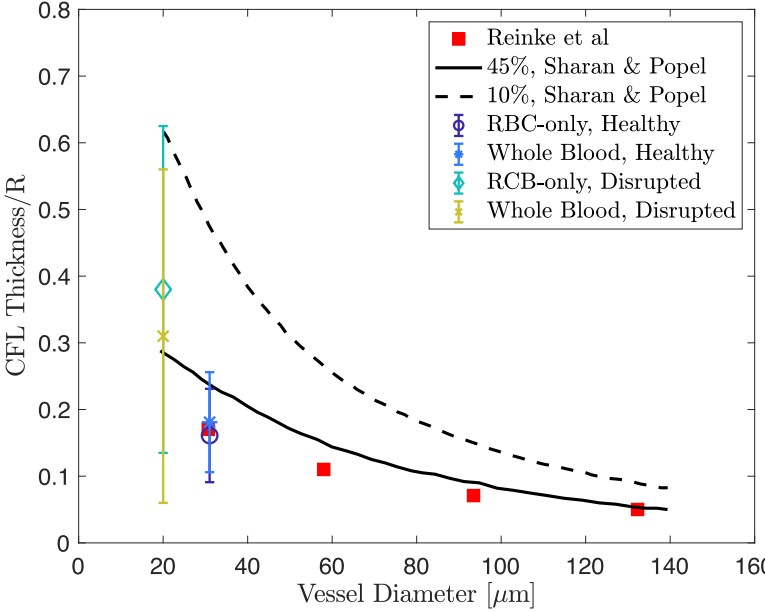

**Fig 18. Dimensionless thickness of the CFL versus vessel diameter and discharge hematocrit.** The black solid and dashed curves represent results extracted from [79]. Red solid squares are experimental data points taken from [12]. Error bars with a circle and an asterisk represent the simulation results in the healthy case for the RBC-only and the whole blood model, respectively. Error bars with a diamond and a cross represent the simulation results in the disrupted case for the RBC-only and the whole blood model respectively. In all four cases, error bars correspond to 95% confidence intervals estimated from the simulation results.

have examined the influence of the the vessel wall geometry induced by changes in the ESL on the wall-induced migration of the RBC. By simulating the wall-induced migration of a single RBC, we find that the vessel wall thickness plays a dominant role in hindering the RBC's ability of moving away from the layer and is independent of the permeability. We note that when the blood vessel wall is thick, it is analogous to a reduced vessel diameter. When the vessel wall is impermeable, it acts as a solid wall. Increasing the vessel wall thickness decreases the vessel diameter, resulting in a decrease in the blood flow and an increase in the resistance in the vicinity of the layer [11, 79, 81]. As a result, it then increases the amount of drag and overpowers the lift, causing the RBC to spend more time moving along the vessel wall instead of moving away from the vessel wall. A similar increasing trend in the drag is observed when the layer is highly permeable. In this case, the migration of the RBC is inhibited and is likely caused by a combined effect of ESL spatial variation as discussed previously in Results and a reduced vessel diameter. On the contrary, the layer's spatial variation has a minimal effect in the healthy condition when the permeability is small. When the ESL is highly permeable, the RBC spends a longer time near the layer, causing the RBC to interact more with the ESL. As a result, the spatial variation of ESL has a more prominent effect on RBC migration in this case.

The amount of drag and lift force experienced by the RBCs together affects the wall-induced migration of RBCs and the formation of the CFL. Our key findings from using the medical images from [43] are that at a fixed hematocrit the CFL thickness is positively correlated to the vessel diameter, whereas for a fixed vessel diameter, the CFL thickness is negatively correlated to the hematocrit. This underscores the potentially important role played by the ESL in affecting the motion of RBCs and the formation of the CFL under various pathological conditions through changing the vessel wall geometry. We note that the current CFL calculation could be affected by the initial positions of cells. Our RBC-only simulations (Fig 13E) predict a few RBCs that become trapped in stagnation zones and spend considerable time there. To more accurately calculate the CFL, further analysis using various initial positions of RBCs and longer simulation time is required. Moreover, our whole blood simulations reveal the non-axisymmetrical nature of the CFL in the healthy case, similar to that observed in experiments [18]. We showed that such non-axisymmetry is also preserved in the case when the ESL is disturbed. However, more comprehensive studies using ESLs during sepsis are required before reaching general conclusions.

Note that in this work we used 2D models to make long-time simulations more computationally feasible. Obtaining the steady-state distributions of RBCs and leukocytes through capillaries requires sufficient statistics and relatively long simulation times, and given the significant computational challenges of simulating fluid-structure interaction in 3D, performing the analogous simulations in 3D requires faster methods and further computational resources. The present study provides good evidence that the qualitative behavior found is robust and likely to persist in 3D. Future studies in 3D are needed to conclusively demonstrate these trends.

In this work, we study the effects of vessel wall geometry caused by changes in the ESL arising during disease on the distribution of RBCs in the vessel. Here we coarse-grained the effects of ESL damage, change in permeability, and blood flow in a narrowed vessels into changes in the vessel geometry. While the ESL has various other physiological roles e.g. interaction with platelets leading to coagulation that are outside the scope of this work, it would be a promising future direction to more comprehensively investigate the influence of the ESL and its varied physiological properties on vessel function. Note that here we have implemented a boundary condition that results in a sinusoidal flow profile, which is different from the parabolic flow profile that would result from prescribing the pressure drop across the channel. Choosing the appropriate boundary condition for an isolated piece of vessel is challenging as it depends on

factors such as upstream and downstream resistances. An interesting future direction would be to investigate the choices of boundary conditions in further detail.

Here we have only discussed the influence of the vessel wall geometry in capillaries far away from the heart and blood flow is essentially steady and laminar. Another interesting future direction would be to consider the scenario in arteries, in which blood flow could occurs at higher Reynolds numbers, and to investigate how ESL changes the vessel wall geometry that lead to changes in the motion of RBCs and the formation of the CFL in the presence of pulsatile flow.

## Supporting information

**S1 Text.** Appendix A. Numerical methods. Appendix B. Convergence study. Appendix C. Validations of blood flow and the membrane elasticity models. Appendix D. Effect of the vessel wall with the RBC represented as a circle. Appendix E. Computational cost for the microscopic ESL and the macroscopic ESL model. Appendix F. The mobility tensor.
(PDF)

**S1 Video. RBC-only simulation with extracted geometry in a healthy condition.**
(MP4)

**S2 Video. RBC-only simulation with extracted geometry during sepsis using the same cell density as in the healthy condition.**
(MP4)

**S3 Video. RBC-only simulation with extracted geometry during sepsis using a lower cell density.**
(MP4)

**S4 Video. Whole blood simulation with extracted geometry in a healthy condition.**
(MP4)

**S5 Video. Whole blood simulation with extracted geometry during sepsis using the same cell density as in the healthy condition.**
(MP4)

**S6 Video. Whole blood simulation with extracted geometry during sepsis using a lower cell density.**
(MP4)

## Acknowledgments

We thank Aleksandar Donev for helpful discussions. All the reported simulations made use of the Brandeis HPCC which is supported by the NSF through DMR-MRSEC 2011846 and OAC-1920147.

## Author Contributions

**Conceptualization:** Ying Zhang, Thomas G. Fai.

**Data curation:** Ying Zhang, Thomas G. Fai.

**Formal analysis:** Ying Zhang, Thomas G. Fai.

**Funding acquisition:** Thomas G. Fai.

**Investigation:** Ying Zhang, Thomas G. Fai.

**Methodology:** Ying Zhang, Thomas G. Fai.

**Resources:** Ying Zhang, Thomas G. Fai.

**Software:** Ying Zhang.

**Supervision:** Thomas G. Fai.

**Validation:** Ying Zhang, Thomas G. Fai.

**Visualization:** Ying Zhang.

**Writing – original draft:** Ying Zhang, Thomas G. Fai.

**Writing – review & editing:** Ying Zhang, Thomas G. Fai.

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
