## [Decision Letter · Decision Letter 0]

7 Nov 2022

Dear Dr Zhang,

Thank you very much for submitting your manuscript "Influence of the endothelial surface layer on the wall-induced migration of red blood cells" for consideration at PLOS Computational Biology.

As with all papers reviewed by the journal, your manuscript was reviewed by members of the editorial board and by several independent reviewers. In light of the reviews (below this email), we would like to invite the resubmission of a significantly-revised version that takes into account the reviewers' comments.

We cannot make any decision about publication until we have seen the revised manuscript and your response to the reviewers' comments. Your revised manuscript is also likely to be sent to reviewers for further evaluation.

Sincerely,

Alison L. Marsden

Academic Editor

PLOS Computational Biology

Lucy Houghton

Staff

PLOS Computational Biology

Reviewer's Responses to Questions

**Comments to the Authors:**

Reviewer #1: The reviewer would like to thank the authors for what appears to be substantial work investigating how the endothelial glycocalyx layer affects red blood cell migration away from vessel walls. The manuscript presents approximately three studies that investigate how the structure and permeability of a protein layer that coats vessels (“endothelial surface layer”, ESL) affects red blood cells as they flow through a vessel. They model the structure by first considering individual protein bundles anchored into the vessel wall with regular spacing in between the bundles (“detailed model”). They then “homogenize” the layer by instead considering a sinusoidal boundary meant to separate the structural components of the ESL from the lumen of the vessel (“coarse-grained model”). An additional aspect of the ESL seems to be included by modeling the fluid within 1 micron (I believe) of the wall as a porous media (ESL’s “fluid component”) to mimic the tendency of the ESL to behave as a porous media. Their 3 studies include 1. A single red blood cell traveling down a tube with a detailed model of the protein/structural components of the ESL, 2. A single red blood cell traveling down a tube with a coarse-grained model of the protein/structural components of the ESL, and 3. Multiple cells traveling through a more physiologically realistic geometry (taken from experimental pictures) using a coarse-grained model of the protein/structural components of the ESL. Throughout they vary the permeability of the fluid component of the model, the density of the proteins, and the thickness of the structural component of the ESL. They come to the conclusions that the ESL’s spatial organization, thickness, and permeability can affect wall-induced migration of red blood cells as they travel through vessels.

To first discuss positives, the authors have made their code available via github. The topic (the effects of ESL properties on the width of the cell free layer, which affects flow resistance in vessels, which affects blood flow and transport in general) does not yet seem to be the subject of others. It is also interesting to note that even though the red blood cells do not directly interact with the structural components of the ESL, the properties of those structural components (particularly thickness) can affect the cell free layer. The attempt to use experimental data was also refreshing to see though the implementation was not completely clear to the reviewer.

While the reviewer appreciates the amount of work put into the paper, there are several concerns that the reviewer was hoping could be addressed. Some of the more major concerns include: A more thorough introduction including motivation and what others have done; A better explanation of the methods used in the study; Using cell position rather than drag and lift as the primary assessment tools during the study; more clearly taking into account multiple potentially confounding factors; and more clearly explaining counterintuitive outcomes. The use of the drag and lift to support results regarding the cell free layer (aka average distance of red blood cells from the wall) rather than more direct red blood cell position estimates make some of these results difficult to believe. For example, the authors state increased drag “hinders the migration of the RBC” towards the centerline even though drag does not push the cell towards or away from the centerline. Drag only acts along the centerline, not towards or away from it.

For the introduction, the authors mention the cell-free layer (CFL) and imply that it is important but never really say why. Related, how might the ESL affect that CFL and related downstream important events? The introduction could also use a better discussion of what’s been done before and how the author’s are presenting a unique contribution. For instance, there is a very similar study by Hariprasad and Secomb that the authors use the data from but they don’t discuss the results, importance, and differences (vs. their study) of that study in the introduction. Another example is the use of 0.5 microns for the outer layer of the endothelial glycocalyx. The reference is from a 2007 Weinbaum paper while other estimates have been made since then (see e.g. works by Fitzroy Curry). In general, it feels like the literature review could and should be a little more extensive.

The reviewer had difficulty understanding how the components of the model fit together and how the study was ultimately organized. It seems like there are structural components modeled either by multiple line segments (“detailed model”) or by a continuous sinusoidal curve (“coarse-grained model”). On top of that is an additional component of the model that turns the fluid into a porous media. I believe this “fluid model” is 1 micron thick thought this seems to be mentioned in just one of the 3 major sections suggesting maybe it changed in the other two studies? Additional comments on the model, particularly inaccuracies and aspects that didn’t seem straightforward, are discussed later.

Most of the results consist of the calculation of drag and lift for these cells. While I think I recall a select few papers considering lift and drag for vesicles (I think) near walls, if the authors are really interested in the cell free layer, why not just plot/analyze the distance of the red blood cell to the wall (e.g. Fig. 3)? Such plots more directly indicate what we can expect in terms of the cell free layer. The drag points directly upstream and, unless the reviewer is missing something, tells us only about what we can expect regarding red blood cell motion upstream and downstream, not towards or away from the centerline of the vessel. I believe integrating the unsteady Stokes equation reveals that the lift is (approximately?) proportional to the average acceleration of the cell (towards the centerline) but it still seems that it is significantly easier to just to plot positions and/or velocities directly. I believe that basing assessments on the more straightforward assessor would be better for readers. I should also note that while lift can be related to the vertical acceleration of the cell, migration of cells (and vesicles) still happens in steady stokes flow where lift (and drag) is zero. Contributing migration solely to lift (e.g. line 13) is probably not a good idea (clarify if the reviewer is misunderstanding).

For confounding factors, it is very difficult to know what factors might be playing an important role in the results presented. It seems the authors always orient the red blood cell 30 degrees. Results could depend on this choice and results are best averaged over all orientations. Alternatively, one could start with a circle to obtain less biased results. I feel I should note that the authors incorrectly state/imply that the proper shape for a 2D red blood cell is a biconcave disk. Biconcave disks only exist in 3D and given past 2D model validations, starting with a circle is arguably just as good and perhaps even better than the biased “biconcave” shape frequently used by 2D modelers. The experimental study seems more like a study regarding a larger vessel vs. a smaller vessel where the smaller vessel has been made smaller not necessarily due to ESL changes but due to inflammatory changes like leukocytes taking up residence at the vessel walls. I imagine that the ESL has been damaged (see, e.g., one of Curry’s responses to a recent paper on a damaged ESL) but the effects of such changes are dwarfed by the changed effective diameter. This latter study doesn’t really seem to be looking at the effects of the ESL, it seems to be looking at the effects of vessel diameter, which has been studied relatively well (as the authors note). In fact, with a bit more simulation I imagine the authors could show that the altered flow doesn’t really depend on the ESL, it depends on the diameter changes induced by inflammatory cells, which may be an interesting result. Another confounding factor is the implementation of uniform flow. It is difficult to tell if this is the right choice for effective boundary conditions. Thicker ESL will increase flow resistance. If we assume constant pressure drop, flow will decrease, and cell migration will be altered in one way. If we instead assume constant flow and cell migration will not change due to decreased flow. I am not sure which of these two is effectively implemented (or what combination) and how that may affect cell migration. It seems that if we increase ESL thickness, the constant body force plus increased vessel resistance will affect the flow but I’m not sure how. This is an additional factor that could affect migration and clarification or use of a more standard boundary condition may be helpful. The red blood cell selection process for the septic case may also be playing a role in outcomes. For the septic case, the selection procedure seems to be biased towards lower percentages of red blood cells which produce wider cell free layers. Either more justification or an alternate procedure that can be justified (based on upstream and downstream expected behaviors) would probably improve the study.

For counterintuitive results, decreasing the vessel width typically enhances lateral migration of red blood cells. Increasing the ESL thickness would, most would expect, effectively decrease the vessel width. Yet the results reported suggest less lateral migration. This result needs further explanation/discussion in terms of why this might happen. Additionally, the fact that this is counterintuitive and why it is should included in such a discussion.

The non-dimensionalization is confusing. The authors state they are using a 1x2 nondimensional domain but later dimensionalized pictures suggest either the impermeable domain has a different aspect ratio than the permeable domain or there is a different effective domain mesh spacing being used between the two, not just one as listed in the parameters table.

Lines 10-14. Red blood cells migrate even in Stokes flow (see above) where lift force is zero. In addition, their migration is believed to be a cause, not an effect (“Upon formation of the CFL, RBCs …migrate away from the wall”), of the cell free layer. I suggest revising these lines.

Lines 18-19. The comment regarding resistance is a fundamental one deserving citation to an authoritative source. Perhaps one of the Pries+Secomb studies might work?

Figure 1: Having a dotted line in a different color for the ESL “fluid model” boundary would be helpful

Equation 6: kb, not ks

Line 95-96: “we … model the ESL as an immobile structure” followed by “we model the layer as an elastic structure”. Immobile sounds rigid, not elastic, is that what is meant by immobile in this context?

Equation 14: How thick is the ESL? Probably should be mentioned here. I found ESL = 1 micron in a later section but not in other sections (see above).

“[0,2]x[0,1] unless otherwise stated”-was this stated differently elsewhere? I feel like this is the only place where dimensionless numbers are used. Maybe just eliminate them (and use standard Navier-Stokes)?

Line 195-96: In general, there is difficulty with reproducibility in this paper. This is one example. I encourage the authors to put this expression in the appendix, for better reproducibility.

Line 199: “at least 5 fibers”. How many exactly? Again, reproducibility.

Lines 208-209: “model stiff fibers”. It sounds like “stiff fibers” correspond to the structural components of the ESL and not the “fibers” that make up the membrane of the RBC. I think it should be stated. It says here that the ESL fibers are tethered but in other places both before (line 98) and after (line 458) that they are free.

Lines 234-237. It appears that increased density increases lift and expected lateral migration/CFL size and that increasing permeability decreases lift and expected lateral migration/CFL size. The way it is stated, it is difficult to tell which trend you are talking about. It feels like you should say each in a separate sentence rather than trying to combine the trends in one sentence?

Figure 5: 95% confidence intervals. It seems like the “four initial conditions” for the cells were used to create these intervals? Probably should mention sooner/in methods section.

Line 287: Description of the experiments and, for instance, how exactly the ESL is estimated is missing from the methods section. True, this has been covered in a different paper, but I think readers would appreciate if this info was in this manuscript (though significantly abbreviated) as well.

Lines 299-300: Not sure what’s trying to be said here. “While we try to extract the ESL boundary for the septic case”?

Line 326: Probably should cite the choice of 6 microns for the radius.

White blood cell study: I believe white blood cell distributions tend to be higher in larger vessels and lower in smaller vessels. The authors should check into this before assuming white blood cells are distributed uniformly throughout the microvasculature.

Figure 10: Doesn’t seem to add much to the discussion. Is there something I’m missing about it?

2D vs 3D model. 3D simulations are feasible nowadays with Karniadakis group making algorithms that can run hundreds of millions of cells and Adrianna Gillman running hundreds of cells in hours (I believe) on her laptop. Developing the algorithms may take some time and 2D simulations may be easier in that respect. 2D simulations are also easier for other reasons as well, but I feel the authors claims regarding computational feasibility for 2D vs 3D are overstated.

Lubrication effect: For tube flows it doesn’t seem like the cells should ever get within 4 grid points of the walls. It also seems like your experimental simulation also didn’t require lubrication elements. Was there another application in mind?

Why unsteady vs steady Stokes flow?

Line 450: What is phi? Again, I would put it here for reproducibility (there are shorter ways to write it than the way Peskin writes it).

Eq. 26: Should say more. Seems to be for small deformations and no reference curvature.

Eq. 27: This doesn’t seem correct. If the cell is far from the origin, the magnitude of the cross-product will always be positive and large giving a very large sum for the “current area” compared to the reference area A0.

Reviewer #2: The authors present computational models of red-blood cells within a permeable endothelial surface layer (ESL) and irregular channel geometries that taken from medical images. Two ESL models are presented; one includes a more detailed brush-like fiber model while the other has a coarse-grained sinusoidal wave geometry. The entire fluid-structure interaction model is formulated using the immersed boundary method. The computational model is used to study the influence of the ESL on microcirculation, wall-induced migration of RBCs and the formation of the cell-free layer observed in vessel flow. Results include a detailed study of the effect of permeability on lift and drag forces in several channel models. The authors find that a larger variation in lift and drag forces when the wall permeability is increased. A similar variation was not observed in simulations where the thickness of the ESL was varied. Simulations with realistic geometries and leukocytes were also included where the authors quantified the cell-free layer and found reduced velocities in the case of a septic ESL. The paper is interesting and well-written. I would like to see some explanation in the variation of lift and drag forces from a fluid-mechanics point of view when the permeability is varied. I found some aspects of the paper that need to be clarified before publication (explained in the comments below).

Major Comments

1. Please provide more biological background and motivation in the introduction. Why is wall-induced migration and maintaining the cell free layer biologically important? Please comment on changes in the ESL during sepsis in the introduction and what causes these changes.

2. What is causes lift forces and wall-induced migration? Is this just a consequence of channel flow (fastest in the middle) that an object moves away from the walls? Please explain.

3. When the layer is more permeable, does the flow become more or less laminar? I wonder if there is a loss in vorticity that is leading to the changes in lift forces. Perhaps the authors could compare the vorticity at different values of permeability, even in the simplified version of the model. It seems like the velocity boundary conditions are no penetration in the normal direction and nonzero in the tangential.

4. In the discussion of Fig. 4 (lines 220 - 240), I found it a bit difficult to see the variation in life and drag as a function of bundle density for each value of permeability. Perhaps the authors could add a figure showing the maximum minus minimum values for the data in Fig. 4 C and E as a function of permeability to highlight the variation in lift and drag (similarly for Figs. 5 - 7).

Also, in this section, if the lift forces are increased, is the cell more likely to remain in the middle of the channel? It would be helpful to see a representative graph of the position of the cell in the channel for a case when the lift forces are low versus high.

Minor Comments

1. Line 30, In the introduction, the authors state that osmotic pressure variation was used to determine the endothelial surface in elastic. How can osmotic pressure be used to determine this? How much deformation occurs in the surface layer?

2. On Line 59, the authors refer to a disrupted scenario, but they haven't defined it at this point in time. Is the disrupted scenario the narrowed channel geometry in Fig. 8E?

3. In the results section, the authors compare their work to [22]. A brief comparison of the present work to previous the modeling approaches such as [22] would be helpful.

4. Line 89 Eq. (7), why is an area conservation term necessary?

5. On p. 2, line 41, the first model refers to the course-grained model, but on p. 4, line 96, the first model refers to the detailed representation of the ESL. Please choose one model as the "first".

6. In both models, does the ESL have internal elasticity or just tether forces? The discussion on pp. 8-10 seems to contradict the description on p. 5 (line 98).

7. On p. 6, line 138, the authors state that they "calculated both forces by averaging over four initial conditions that either uniformly sample the distance between two neighboring bundles for the detailed model or one wavelength of the sine wave for the coarse-grained model." How are these 4 initial conditions different? I had a difficult time determining this from the text. Was the orientation of the cell fixed at 30 degrees in each case? Please clarify.

8. On p. 8, line 186, is the biconcave shape just an initial condition? Did the authors add a reference curvature or use a different energy functional maintain this shape? Please explain.

9. On p. 10, line 248, was the height of the bundles increased to model a thicker ESL? Please clarify.

10. On p. 15, Fig. 8, which value of permeability was used for these simulations? Table 3 shows a range of values.

11. I am confused about the different between healthy and septic with the extracted geometry. Is the main difference the narrowing of the channel? It seems that the ESL is not modeled as a permeable structure for the whole blood simulations. Does the ESL act as an elastic no-slip boundary?

12. On p. 14, line 317, how exactly is the cell free layer measured? Is it an averaged distance from the red curve to where the steady state distribution is nonzero in Fig. 8 and 9? Please explain.

13. On p. 16, line 333, can the authors comment on why the RBCs move slower in the disrupted case? Is the background channel flow the same in healthy versus disrupted?

14. For the study described on p. 16, last paragraph, do the authors use the extracted geometry when varying the vessel diameter? Please explain.

**Have the authors made all data and (if applicable) computational code underlying the findings in their manuscript fully available?**

Reviewer #1: Yes

Reviewer #2: Yes

PLOS authors have the option to publish the peer review history of their article (what does this mean?). If published, this will include your full peer review and any attached files.

Reviewer #1: No

Reviewer #2: No
---

## [Decision Letter · Decision Letter 1]

1 Feb 2023

Dear Dr Zhang,

Thank you very much for submitting your manuscript "Influence of the endothelial surface layer on the wall-induced migration of red blood cells" for consideration at PLOS Computational Biology.

As with all papers reviewed by the journal, your manuscript was reviewed by members of the editorial board and by several independent reviewers. In light of the reviews (below this email), we would like to invite the resubmission of a significantly-revised version that takes into account the reviewers' comments.

We cannot make any decision about publication until we have seen the revised manuscript and your response to the reviewers' comments. Your revised manuscript is also likely to be sent to reviewers for further evaluation.

Sincerely,

Alison L. Marsden

Academic Editor

PLOS Computational Biology

Lucy Houghton

Staff

PLOS Computational Biology

Reviewer's Responses to Questions

**Comments to the Authors:**

Reviewer #1: The reviewer thanks the authors for their work on the most recent manuscript. The reviewer appreciates the several clarifications that have been made and still appreciates the availability of their work, its scope, and the time and effort that it took. Still, the reviewer is concerned that many suggested changes were only minimally addressed and/or made in the manuscript and encourages the authors to look over the reviewer's previous suggestions more carefully/again.

One major concern is that there are multiple implied statements that the authors seem to make that are not exactly canonical. One example, which I already commented on previously, is that the in vivo study implies to the naive reader that the ESL thickness actually increases to 6-10 microns because during sepsis because the ESL entrains/entraps multiple immune cells. The studies I have read state that the ESL is considered to be a layer of various molecules and they never include cells in the layer. One sentence is included on how disrupting the ESL is believed to bring in immune cells and cause narrowed vessels. The rest of the manuscript, however, reads as if disruption increases thickness and narrows the vessel without any reference to the immune cell role. Another similar worrying statement is that decreased CFL leads to increased blood velocity. This is entirely against canon as the CFL (I'm fairly certain) was originally hypothesized to explain why blood was going faster than expected, a wide CFL was needed to explain faster, not slower, velocities. Related, we have an extreme change in hematocrit from 45% to 10%. Based on this alone, it is no surprise to me that the CFL is much wider in the low hematocrit scenario than in the other.

These issues all stem back to confounding factors. When multiple things change at once, it is difficult to know which is responsible for the effects seen and can lead to false (implied and/or perceived) claims such as how the ESL gets microns thicker during sepsis. I encourage the authors to try to avoid changing everything at once, as they do in their last study, and instead develop a more systematic study that changes things one at a time. E.g. with/without white blood cells, with same vs different hematocrit, with same/different velocities, with same/different widths, etc. Saying that "lateral migration is a result of competing effects", while true, does not help us identify which effects are more or less important.

What follows are general, more minor comments.

Please define "mobility". What is the actual tensor? (Perhaps in supplementary info.)

To make the macro model more obvious, one should probably name and label in Figure S5 the fibers for both micro and macro models (e.g. 9-10 does a little bit of that).

For equation 6, I'm guessing the energy is meant to be curvature squared as it is in the Helfrich bending energy with zero reference curvature. If that's the case, I'm pretty sure the curvature expression used in equation 6 is for small deformations (see Stewart Calculus, for instance). When q corresponds to the current arc length, the expression yields the curvature but if q corresponds to Lagrangian coordinates that can be significantly stretched and differ from the geometric arc length, the expression does not yield the exact curvature.

There are several typos/grammar issues throughout. Unfortunately, there seem to be more in this version of the manuscript than there were previously. Fixing this, I expect, will make the manuscript read more smoothly.

There seem to be introductory paragraphs in some of the Results subsections with material that probably belongs in the introduction.

The hematocrits chosen. If one has information about the expected hematocrit as a function of vessel size, one should use that to help design one's study. If your healthy case should expect 35% hematocrit while the unhealthy case should expect 20%, then why not use those percentages (while also perhaps taking steps on the way, e.g. comparing 35% in both and comparing 20% in both before comparing 35 vs 20).

The authors have not yet made it clear why drag that goes upstream/downstream only might affect lateral migration. The efforts so far have not been sufficient.

"due to the stiffness of ESLs" Is permeability or density meant in the place of "stiffness". I don't recall reading about how ESL stiffness affects flow.

Equation 28 is still incorrect as X_i^0 is not explained sufficiently. Why is the 0 there? Perhaps the authors mean "position vector relative to the cell centroid"? By the way, the formula is also problematic/doesn't work for cells that deform too much and lack a star-point. It may or may not matter for forces, but the energy will be off.

As always, if some of my claims seem off, please clarify the issues so I can better understand the claims the authors are making.

Reviewer #2: The authors made substantial revisions and addressed my comments. I recommend publication after addressing the following minor comments.

On p. 3, line 59, please provide more description of the ESL model from [50]. In particular, the endothelial glycocalyx (EG) appears to be modeled as a chain with elasticity and bending rigidity.

In Fig. 11 A, the text should be "variance with varying spatial frequency", not bundle density.

In the new S4 Appendix, the figure references are incorrect. On line 603, I believe S4 Fig should be S6 Fig. On Line 604, S6 Fig should be S8 Fig. Please carefully check the figure references in this section.

On p. 29, the x-axis should be density of bundles to be consistent with Fig. 3.

**Have the authors made all data and (if applicable) computational code underlying the findings in their manuscript fully available?**

Reviewer #1: Yes

Reviewer #2: Yes

PLOS authors have the option to publish the peer review history of their article (what does this mean?). If published, this will include your full peer review and any attached files.

Reviewer #1: No

Reviewer #2: No
---

## [Decision Letter · Decision Letter 2]

17 Apr 2023

Dear Dr Zhang,

Thank you very much for submitting your manuscript "Influence of the vessel wall geometry on the wall-induced migration of red blood cells" for consideration at PLOS Computational Biology. As with all papers reviewed by the journal, your manuscript was reviewed by members of the editorial board and by several independent reviewers. The reviewers appreciated the attention to an important topic. Based on the reviews, we are likely to accept this manuscript for publication, providing that you modify the manuscript according to the review recommendations.

Sincerely,

Alison L. Marsden

Academic Editor

PLOS Computational Biology

Lucy Houghton

Staff

PLOS Computational Biology

Reviewer's Responses to Questions

**Comments to the Authors:**

Reviewer #1: I appreciate the great deal of work the authors have put into their manuscript including more thorough consideration of many of my comments. I think the manuscript has greatly improved as a result. While I appreciate the more methodical approach that has been adopted by changing factors more strategically, there are a few smaller issues that I am still unclear about.

The authors comment that “drag toward the no-slip wall” is somehow important for the migration process. This comment and the use of total drag as an indicator for formation of the CFL still confuses me. I feel I need to try my point one more time so that hopefully it’s clear to the authors and they are able to address it better. Drag operates up/downstream, not towards the no-slip wall. My understanding is that drag (especially the one plotted in the paper) is the total force operating up/downstream on the cell while lift is the total force operating to/away from the wall. If this is the case (correct me if you are using a different definition of drag), I am still not clear on why total drag continues to be used as some sort of indicator for CFL enhancement and thickness. Is the argument that drag causes asymmetric deformations which causes migration? If so, I’m pretty sure the situation is similar to the story about pressure in incompressible flow, raising or lowering the pressure by a constant does not affect dynamics whatsoever. Only changes in pressure gradient matter. Similarly, changes in the total drag doesn’t matter in terms of deformation of cells. Only variations in the upstream/downstream stress distribution across the cell matter for morphological changes. For instance, often there is upstream stress on the portion of the cell near the wall and downstream stress on the portion of the cell away from the wall, causing the cell to deform into slipper-like shapes that move away from the wall. I believe these sort of asymmetric effects is what the cited references discuss. The total drag in such scenarios could be any value, all that matters is the differences in local drag from top to bottom in the cell. If the total drag is, in fact, somehow an indicator of the magnitude of such variation, it should be explained better how that is the case. It doesn’t seem to be clear from the references given as they site morphological asymmetries that arise (which I don’t think can arise, as argued above, due to total drag on the particle). Both references also seem to consider adhesion which may be giving rise to “insights” not applicable to non-adhering cells. E.g. that adherence mediated “drag” gives rise to more deformation and higher migration. I encourage the authors to consider this once more, especially since the use of plots of total drag is so prevalent throughout the text, and either make their case stronger than it currently is or to reevaluate.

The implementation of permeability still is unclear to me. In particular, is the permeability implemented by applying equations 15 and 16 only at the red locations in Figure 1? Are they also applied at the blue locations in Figure 1? If they are applied at the red fibers, particularly in 1b, it seems like changing bundle density also affects permeability. Is this correct? Clarification on permeability vs. bundle density (vs. thickness) could probably be helpful.

More minor comments/suggestions

In the abstract, “formation of a cell-free layer” should probably read “enhancement of the cell-free layer” or something similar. There is always a cell-free layer, just its size may be altered.

“which potentially leads to a heterogeneous distribution of RBCs”. Again, RBCs are typically heterogeneously distributed and I suggest “which can increase the heterogeneity of the RBC distribution in capillaries” or something similar.

In my experience, the Helfrich-type bending energies are frequently used but implementation details are seldomly explained leading to difficulties for those trying to follow the most “accepted” approach. For instance, very few papers explain whether the integration of the Helfrich energy density is over the original configuration (Lagrangian) or over the current configuration (Eulerian). For instance, a paper by Strychalski et al. suggests Helfrich’s energy density is typically integrated over the deformed configuration. This is why I encourage statement of assumptions within the paper rather than simply referring to other papers, especially when practices in those papers may disagree from practices in other papers supposedly implementing the same thing. I am glad to hear the difference is less than 3% but I hope the authors understand why I encourage them to more clearly state assumptions, when possible.

It's good to see that equation 28 has been fixed. Ultimately, sum(magnitude(x_i)) <> magnitude(sum(x_i)). The fixed formula does not make that past persistent mistake.

“although Eq 6” should probably be “although partial^2X/partial^2”. In addition, the rest of the sentence is awkward. Perhaps something like although this does not yield the exact curvature, especially for large deformations, here it differs from the exact curvature by less than 3%.

181-182. Are tether points at the red locations? The blue locations too? I suppose I was unclear on the blue location in Figure 1. Are those explicitly tracked Lagrangian points or an artist rendering choice?

283-4 RBC [migration] is significantly inhibited…caused by a decrease in the drag and an increase in the lift. Increased lift means more migration. I’m not sure why the authors say migration is inhibited in such a case.

RCB vs RBC. Centered of mass vs. center of mass.

When adding the leukocytes, the authors see the same behavior. Ideally there is at least some difference (e.g., less of a decrease in both quantities) that could be highlighted (or else it would have been sufficient to just show just one of the two cases). Even if there is no {/it significant} difference, that is a result worth stating.

There are many statements of the sort “they are the same”. This should typically be accompanied by more specifics like, “to within 1%”. This is a good practice in general.

Discussion of limitations of study should include discussion of boundary conditions. The “correct” boundary conditions for an isolated vessel in a region of sepsis is very difficult to ascertain. It depends on upstream and downstream resistances and comparing healthy and septic scenarios is very difficult because of the uncertainty of those boundary conditions. Depending on the situation, the septic region may receive a significantly decreased or a significantly increased amount of flow. The approach used here is, in addition, somewhat problematic as it is not clear if the approach corresponds to a prescribed velocity or to a prescribed pressure drop, which tend to be the two most well-accepted set of boundary conditions for tube flow. As such, I’m not really sure what this means for where these simulations lie between the expected extremes that might occur in vivo. A discussion of all of this would be appropriate.

Reviewer #2: I recommend acceptance.

**Have the authors made all data and (if applicable) computational code underlying the findings in their manuscript fully available?**

Reviewer #1: Yes

Reviewer #2: Yes

PLOS authors have the option to publish the peer review history of their article (what does this mean?). If published, this will include your full peer review and any attached files.

Reviewer #1: No

Reviewer #2: No

Figure Files:

Data Requirements:

Reproducibility:

References:

---

## [Decision Letter · Decision Letter 3]

3 Jun 2023

Dear Dr Zhang,

We are pleased to inform you that your manuscript 'Influence of the vessel wall geometry on the wall-induced migration of red blood cells' has been provisionally accepted for publication in PLOS Computational Biology.

Best regards,

Alison L. Marsden

Academic Editor

PLOS Computational Biology

Lucy Houghton

Staff

PLOS Computational Biology

Given that we have already gone through several rounds of revisions on this, I recommend acceptance at this point. Please note that one reviewer still has raised considerations that should be taken into account. I would like to see the authors address the comments of the reviewers in the final uploaded version (both grammatical points and also more substantial points raised).

Reviewer's Responses to Questions

**Comments to the Authors:**

Reviewer #1: The reviewer once again thanks the authors for their work and addressing some of the concerns the reviewer had. The reviewer also appreciates the significant amount of work done on the supplementary materials. Still, the reviewer still has several comments regarding the material. Please see them below.

Major issues

The total drag issue is still not clear. The closest the manuscript seems to get towards understanding why total drag might be useful is to say that higher total drag and lower lift makes it so that the cell stays closer to the wall for longer. At the same time, this is counterintuitive as higher {/it local} drag that arises when cells are nearer to the wall (with O(normal) drag still on the side of the cell away from the wall) causes more deformation and more deformation tends to be associated with more migration away from a wall (for instance, rigid spheres in low Reynolds number flow cannot migrate) and larger, not smaller, cell free layers. A clear discussion of the role of total drag and lift and how they affect migration and the cell free layer belongs as early as possible, e.g. in the introduction. The additions (e.g. lines 101-108) made are fairly ambiguous and not very clear. E.g., “force causes motion”. This is true but doesn’t tell us about total drag’s specific role and why graphs of it are placed on most pages. The importance of the total lift is briefly mentioned here (and is believable), but not the total drag.

Figure 2: Why are some of the streamlines going backwards? I believe most CFD practitioners would find this unacceptable. Either adjust methods or discuss sufficiently.

Minor comments:

Abstract: On first appearance I suggest “endothelial lining” -> “endothelial lining of endothelial surface layer (ESL)” and then using ESL thereafter. As is, it is not immediately clear that “endothelial lining” and “endothelial surface layer” are effectively the same thing (or at least close enough to the same thing).

“various pathological conditions” -> “various pathological conditions, such as sepsis, “. Basically, I suggest the term “sepsis” should appear in the abstract, which is where you got your experimental data from.

Summary: “induced by ESL”-> “induced by ESL changes”

Intro: Fahraeus effect is mentioned without definition. That definition is needed for non-experts (that are still experts in compbio/cfd).

“In a more recent study…dissipative particle dynamics” References are usually placed at the location at which a study is first mentioned, not after the description of the study is done.

Missing reference: Triebold, 2022

Again, in general, grammar. E.g. “motion of a single RBC vary” vs. “motion of a single RBC varies”. Other examples appear throughout.

Methods:

144-45 Blood is non-Newtonian. The plasma can be/is often considered Newtonian.

“In all simulations ka is picked” -> “ka is picked so that in all simulations”. The former makes it sounds like multiple values of ka may be used.

“diffusing with the blood flow”->”advecting with the blood flow”? Not sure how an elastic structure diffuses with the flow.

“bundles of fibers…defined identically as Eq 5”->So, no bending elasticity? Just thought I should check.

Figure 1: “Dashed line”. I apologize for my confusion with the blue dashed line. When I first made the comment, I had two issues in mind, 1) The fluid domain might actually be smaller than the 2piX4pi box shown and 2) IBM people sometimes use oversized fluid domains and model walls within those domains use stiff tethered fibers (e.g. see battista’s IB2d examples). Also coming in to play is that some people would consider the blue dashed line the “vessel wall” while the endothelial structures extend from that wall into (and are immersed in) the fluid. Reflecting on all three of these things, I think careful readers will be ok (e.g. “vessel wall” <> blue dashed line in your work).

Equations 15-16. Why aren’t these combined like 13 and 14?

“The model is spatially discretized as in 67”. Authors should include something like “finite differences” or “fast fourier transforms”, at least some indication of what method was used (then interested readers can go to the appendix as mentioned).

Results:

Variable h redefined to be wall thickness instead of top portion of the domain (h in subscript earlier). Avoid redefinitions.

Figure 2: There are 4 subplots here that should be labeled A, B, C, and D.

Figure 4: “the distance of the RBC to the wall increases more with a thicker vessel wall” -> “the distance of the RBC to the wall increases more with a thinner vessel wall”. The change in the black curves (thinner) is more than the change in the red curves (thicker).

Figure 8: Weird bump at spatial frequency of 2 should be explained if not already.

Looking up the paper where the “boundary condition” came from, it is not equivalent to a constant pressure drop as it yields a sine flow profile instead of the standard parabolic flow profile. This should be included in your discussion.

Reviewer #2: I recommend acceptance.

**Have the authors made all data and (if applicable) computational code underlying the findings in their manuscript fully available?**

Reviewer #1: Yes

Reviewer #2: Yes

PLOS authors have the option to publish the peer review history of their article (what does this mean?). If published, this will include your full peer review and any attached files.

Reviewer #1: No

Reviewer #2: No

---

## [Editor Report · Acceptance letter]

10 Jul 2023

PCOMPBIOL-D-22-01430R3 

Influence of the vessel wall geometry on the wall-induced migration of red blood cells

Dear Dr Zhang,

I am pleased to inform you that your manuscript has been formally accepted for publication in PLOS Computational Biology. Your manuscript is now with our production department and you will be notified of the publication date in due course.

With kind regards,

Zsofi Zombor
